



# Flow separation, dipole formation and water exchange through tidal straits

Ole Anders Nøst[1,3] and Eli Børve[2,4]

[1]Akvaplan-niva AS, 7462 Trondheim, Norway
[2]The University of Oslo, Department of Geosciences, 0315 Oslo, Norway
[3]Nord University, 8026 Bodø, Norway
[4]Akvaplan-niva AS, 9296 Tromsø, Norway

**Correspondence:** Ole Anders Nøst (oan@akvaplan.niva.no)

**Abstract.** We investigate the formation and evolution of dipole vortices and their contribution to water exchange through idealized tidal straits. Self-propagating dipoles are important for transporting and exchanging water properties through straits and inlets in coastal regions. In order to obtain a robust data-set to evaluate flow separation, dipole formation and evolution and the effect on water exchange, we conduct 164 numerical simulations, varying the width and length of the straits as well as the tidal

forcing. We show that dipoles are formed and start propagating at the time of flow separation, and their vorticity originates in the velocity front formed by the separation. We find that the dipole propagation velocity is proportional to the tidal velocity amplitude, and twice as large as the dipole velocity derived for a dipole consisting of two point vortices. We analyse the processes creating a net water exchange through the straits and derive a kinematic model dependent on dimensionless parameters representing strait length, dipole travel distance and dipole size. The net tracer transport resulting from the kinematic model

agrees closely with the numerical simulations and provide understanding of the processes controlling net water exchange.

## 1 Introduction

Knowledge of coastal ocean transport processes is vital for predicting human impact on the coastal marine environment. Coastal industry discharges pollutants and nutrients into the ocean, and, in order to understand the impact on the environment, we need coastal ocean circulation models to calculate concentrations and pathways of spreading. Setting up such models for a complex coastline requires a high level of understanding of near-shore transport processes in order to realistically represent these in the

models. In shallow coastal regions with complex topography, tides are often a dominant driver of the ocean circulation and transport. In this study, we investigate the exchange process of tidal pumping through narrow tidal straits.

Tidal pumping is an important mechanism responsible for transport of water properties and particles like fish eggs, nutrients, and pollution between estuaries and the open ocean, or in coastal regions with complex geometry in general (Amoroso and

Gagliardini, 2010; Ford et al., 2010; Chadwick and Largier, 1999; Fujiwara et al., 1994; Brown et al., 2000). The exchange process results from an asymmetry of the flow field between the ebb and flood phase of the tide (Stommel and Farmer, 1952; Wells and van Heijst, 2003). The flow asymmetry occurs when the tidal current interacts with a topographic constriction like a strait or an inlet. The flow speeds up when entering the constriction in order to conserve volume. The sea surface in the



constriction is lowered as potential energy is converted to kinetic energy when the flow accelerates towards the constriction.
Downstream of the constriction, the cross-sectional area increases and the sea surface rises. Here, both friction and pressure
forces work to decelerate the flow, which is a necessary condition for flow separation (Kundu, 1990). The friction is strongest
in the frictional boundary layer near the shore, and because the friction and pressure both work in the same direction, the
flow is likely to come to a halt close to the shoreline somewhere downstream of the constriction. When this happens, the flow
separates from the coastline (Kundu, 1990; Signell and Geyer, 1991). The flow separation leads to the asymmetry between the
tidal currents entering and exiting the strait.

The water exchange process driven by flow asymmetry, was first pointed out by Stommel and Farmer (1952). Later, the
studies by Kashiwai (1984a) and Wells and van Heijst (2003) have shown that the water exchange is closely related to formation
of self-propagating dipoles. When the flow separates from the coast, a vortex forms at the point of separation. If the flow
separates at both sides of the exit, two vortices of opposite sign will form with a separation distance roughly equal to the width
of the strait. Dependent on the strength of the vortices and the distance between them, the two vortices may interact and form
a self-propagating dipole. The dipole will be drawn back into the strait or continue moving away and escape the return flow,
depending on the dipole location and propagation speed when the flow reverses. Since the dipole is filled with waters ejected
from the strait, it will contribute to a considerable water exchange if it escapes the return flow.

The dipole position at the time of flow reversal depends on the dipole velocity. The propagation of dipoles has been studied
for more than 100 years (Lamb, 1916; Batchelor, 1967; Kundu, 1990), and the velocity of a self propagating dipole is typically
represented as

$$U_{dip} = \frac{\Gamma}{2\pi b}. \tag{1}$$

Here $b$ is the distance between the vortex centers, and $\Gamma$ is the magnitude of the circulation in each of the two vortices, assuming
they are of equal strength. Equation 1 is valid as long as the distance between the two vortices is large compared to their core
radius (Yehoshua and Seifert, 2013; Delbende and Rossi, 2009; Habibah et al., 2018). Habibah et al. (2018) show that a
correction to the velocity given by Eq. 1 occurs in the 5th order of $a/b$ where $a$ is the core radius of the vortices. In cases where
$a/b$ increases, the vortices becomes elliptical and the dipole propagation velocity decreases (Delbende and Rossi, 2009).

Equation 1 describes the propagation velocity of a dipole moving by self-propagation in an otherwise non-moving ocean.
It is unclear whether this is valid for a dipole formed in a tidal strait, where the background flow is clearly not zero. Also,
dipoles propagating away from the strait often remain attached to the strait via a trailing jet, which provides a pathway of
mass, momentum and vorticity from the strait into the dipole (Wells and van Heijst, 2003; Afanasyev, 2006). As the dipole
accumulate vorticity, the circulation in the dipole increases, and the propagation velocity should therefore accelerate according
to Eq. 1. However, this is not necessarily true. In a lab experiment investigating dipole formation by a steady channel jet,
Afanasyev (2006) found that the dipole propagated with constant speed, even though the dipole continuously accumulated
vorticity fed by a trailing jet.

The circulation of the dipole vortices is an important parameter, and to determine the circulation it is vital to know the source
of vorticity. A much used assumption is that the vorticity is created in the viscous boundary layer (Wells and van Heijst, 2003;





Nicolau del Roure et al., 2009; Bryant et al., 2012). Another possible source is the flow discontinuity resulting when the flow separates from the coastline (Kashiwai, 1984a, b). Kashiwai (1984a, b) and Wells and van Heijst (2003) both assume that all
vorticity generated in the strait accumulates in the dipole vortices. The circulation can then be expressed as $\Gamma \propto U^2 T$, where $T$ is the tidal period and $U$ is a characteristic velocity scale for the strait (Kashiwai, 1984b; Wells and van Heijst, 2003). However, Afanasyev (2006) showed that the vorticity is divided between the dipole vortices and the jet trailing the dipole. In addition, Afanasyev (2006) introduced a new time-scale, which he called the "startup time", $t_s$. The startup time indicates the moment when the dipole starts translating after an initial period of growth, where the jet is injected into the dipole.

The net tracer transport through a tidal strait is commonly classified by the nondimensional Strouhal number, $S_t$, defined as (Kashiwai, 1984a; Wells and van Heijst, 2003; Nicolau del Roure et al., 2009)

$$S_t = \frac{W}{UT}, \tag{2}$$

where $W$ is the strait width, $T$ is the tidal period and $U$ is the velocity scale characterising the velocity in the strait. $W$ can also be seen as a characteristic spatial scale of a dipole formed at the strait exit, and in this case $S_t$ is a measure of the ratio
between linear and non-linear acceleration terms. The center of the dipole vortices are pressure minimums, and the non-linear acceleration associated with the azimuthal velocity of the vortices is balanced by pressure forces. Thus, for a dipole vortex to exist, $S_t \ll 1$ is a necessary condition.

Net tracer transport by tidal pumping is associated with $S_t < S_{tc}$, where $S_{tc}$ is a threshold value of $S_t$ (Kashiwai, 1984a; Wells and van Heijst, 2003). The threshold value of the Strouhal number arrives from a kinematic consideration of the dipole
movement over one tidal period, and separates between dipoles who escape the return flow and the dipoles that returns to the strait during the subsequent phase of the tide (Kashiwai, 1984a; Wells and van Heijst, 2003). Dipoles escaping the return flow contribute to net water exchange through the strait.

In this study, our aim is to understand how the geometric constraint of a tidal strait influences the effectivity of tidal pumping, and the universality of the commonly used Strouhal number. We systematically perform 164 numerical simulations in an
idealized tidal strait, varying the width and length of the straits as well as the amplitude of the tidal forcing. The results of the simulations are analysed with focus on flow separation, dipole formation and propagation and net water exchange. Finally, we derive a simple kinematic model for net tracer transport that fits well to the results from the simulations and brings understanding to the process of water exchange through a tidal strait.

## 2 Modelling

### 2.1 The model

We use the Finite Volume Community Ocean Model (FVCOM) (Chen et al., 2003). FVCOM has been used in numerous studies of coastal and estuarine waters (Li et al., 2018; Lai et al., 2015, 2016; Sun et al., 2016; Chen et al., 2021) and also globally and in the Arctic Ocean (Chen et al., 2016; Zhang et al., 2016). FVCOM uses an unstructured triangular grid in the horizontal and terrain-following $\sigma$-coordinates in the vertical (Chen et al., 2003). The model solves the equations for momentum and mass





conservation as well as the equations for temperature, salinity and density. In our case, we set temperature, salinity and density
to constant values and FVCOM then solves the following equations

$$\frac{\partial u}{\partial t} + u\frac{\partial u}{\partial x} + v\frac{\partial u}{\partial y} + w\frac{\partial u}{\partial z} - fv = -\frac{1}{\rho_0}\frac{\partial p}{\partial x} + \frac{\partial}{\partial z}\left(K_m\frac{\partial u}{\partial z}\right) + F_u$$

$$\frac{\partial v}{\partial t} + u\frac{\partial v}{\partial x} + v\frac{\partial v}{\partial y} + w\frac{\partial v}{\partial z} + fu = -\frac{1}{\rho_0}\frac{\partial p}{\partial y} + \frac{\partial}{\partial z}\left(K_m\frac{\partial v}{\partial z}\right) + F_v \tag{3}$$

$$\frac{\partial u}{\partial x} + \frac{\partial v}{\partial y} + \frac{\partial w}{\partial z} = 0$$

$$\frac{\partial p}{\partial z} = -\rho_0 g.$$

$x$, $y$ and $z$ are the Cartesian coordinates in east, north and vertical directions, respectively; $u$, $v$ and $w$ are the $x$, $y$ and $z$ components of velocity, respectively; $p$ is pressure; $\rho_0$ is the constant density; $f$ is the Coriolis parameter; $g$ is the acceleration

of gravity; $K_m$ is the eddy diffusion coefficient and $F_u$ and $F_v$ are the diffusion terms for horizontal momentum in $x$ and $y$ directions, respectively. The calculation of $K_m$ is done with the Mellor and Yamada (1982) level 2.5 turbulent closure scheme, modified by Galperin et al. (1988). $F_u$ and $F_v$ are calculated using the eddy parameterization method by Smagorinsky (1963). The diffusion coefficient within $F_u$ and $F_v$ is given by

$$A_m = 0.5C\Omega\sqrt{(\frac{\partial u}{\partial x})^2 + 0.5(\frac{\partial v}{\partial x} + \frac{\partial u}{\partial y})^2 + (\frac{\partial v}{\partial y})^2}, \tag{4}$$

where $C$ is a constant, set to 0.1 in our case, and $\Omega$ is the grid cell area.

The surface boundary conditions are given by

$$\left. \begin{array}{c} K_m\left(\frac{\partial u}{\partial z}, \frac{\partial v}{\partial z}\right) = \frac{1}{\rho_0}(\tau_{sx}, \tau_{sy}) \\ w = \frac{\partial \zeta}{\partial t} + u\frac{\partial \zeta}{\partial x} + v\frac{\partial \zeta}{\partial y} \end{array} \right| z = \zeta(x,y,t), \tag{5}$$

where $\tau_{sx}$ and $\tau_{sy}$ are the surface stress in $x$ and $y$ directions, respectively, and $\zeta$ is the surface elevation. The bottom boundary conditions are given by


$$\left. \begin{array}{c} K_m\left(\frac{\partial u}{\partial z}, \frac{\partial v}{\partial z}\right) = \frac{1}{\rho_0}(\tau_{bx}, \tau_{by}) \\ w = -u\frac{\partial H}{\partial x} - v\frac{\partial H}{\partial y} \end{array} \right| z = -H(x,y), \tag{6}$$

where $\tau_{bx}$ and $\tau_{by}$ are the bottom stresses in the $x$ and $y$ direction, respectively and $H$ is the bottom depth. The bottom stresses are given by

$$(\tau_{bx}, \tau_{by}) = C_d\sqrt{u^2 + v^2}(u,v), \tag{7}$$

where the drag coefficient $C_d$ is given by

$$C_d = max\left(\frac{\kappa^2}{ln(\frac{z_b}{z_0})^2}, 0.0025\right). \tag{8}$$

Here, $\kappa$ is von Karmans constant ($\sim 0.4$), $z_0$ is the bottom roughness set to be 0.001 m and $z_b$ is height above bottom of the lowest horizontal velocity level.




## 2.2 Setup of simulations

The model domain is bounded by a half-circled open ocean and a straight coastline on the eastern side (Figure 1). The full
domain is 500 km in the north-south direction and up to 250 km in the east-west direction. At the center of the eastern
boundary, we place a peninsula and an island separated by a strait. It is this strait that is the focus of our study. The idea behind
this configuration is that the pressure difference over the length of the strait will be set by the tidal wave travelling in the open
ocean and not by the flow through the strait. In this way, the flow through different strait geometries will be forced similarly.

Surface stress (Eq. 5) is set to zero, and the only forcing of the simulations is a northward propagating Kelvin wave specified
at the half-circled western boundary

$$\zeta_{obc} = A_t e^{\frac{(x-x_c)}{R_d}} sin(ky - \omega t). \tag{9}$$

Here, $\omega = 2\pi/T$, $T$ is the M2 tidal period (12.42 hours), $k = \omega/\sqrt{gH}$, $x_c$ is the constant position along the x-axis of the
straight eastern coast (ignoring the peninsula) and $R_d$ is the Rossby radius of deformation. Equation 9 describes a classical
Kelvin wave moving northwards with the coast to the right (Gill, 1982). The Coriolis parameter corresponds 70°N latitude and
the depth $H = 100$ m, giving a Rossby radius $R_d \simeq 230$ km. The surface elevation given by Eq. 9 is specified at the boundary
nodes. The velocities in FVCOM is located in the center of each triangular cell, and not directly at the boundary. The velocities
in the open boundary cells are calculated based on the assumption of mass conservation (Chen et al., 2003, 2011).

In order to investigate the geometric effects on the tidal pumping, we vary the width of the strait, W, from 1 km to 12 km,
and the length of the strait, L, from 4 km to 22 km. The curvature of the coastline at the strait entrance and exit is equal and
shaped as a quarter of a circle with a radius of $R = 2$ km. The strait is directed north-south, and the geometry and coordinates
used in the study is shown in Figure 2b. In total we conduct 164 idealized simulations using 82 different strait geometries and
two different amplitudes of the tidal forcing ($A_t = 1$ and $A_t = 0.5$, see Eq. 9).

We simulate a homogeneous ocean over a flat bottom of 100 m depth. To avoid unwanted effects of boundary layers near a
vertical wall, we use a sloping bottom at the innermost 600 m from the coastline inside the strait (Figure 2a). The minimum
depth is 5m. The water column is divided into two layers in the vertical. $z_b$ is roughly equal to a quarter of the total depth
resulting in a slightly increased drag coefficient over the shallow depth near the sides of the strait (Eq. 8). $z_b = 1$m gives
$C_d = 0.0034$ while $C_d = 0.0025$ for $z_b > 2.8$m.

Inside the strait, the resolution is 50 m along the coastline. Inside the focus region surrounding the strait, the resolution
linearly coarsens to 200 m with distance from the coast. The focus region is, in addition to the strait itself, the half circle (of
radius $= W/2 + 2R$) of high resolution at both sides of the strait entrances (Figure 1). Outside the focus area, the resolution
coarsens further to 2 km both at the western tip of the island and at the coastline to the east. At the western open boundary the
resolution is 20 km.

The simulations are run for a total of 20 days. First, a 10 days spin-up, before we introduce a passive tracer, which is simulated
using the Framework for Aquatic Biogeochemical Models (Bruggeman and Bolding, 2014, FABM) coupled to FVCOM. The
initial concentration of the tracer is set to 1 $m^{-3}$ inside a rectangular box south of the strait, and 0 $m^{-3}$ elsewhere (left



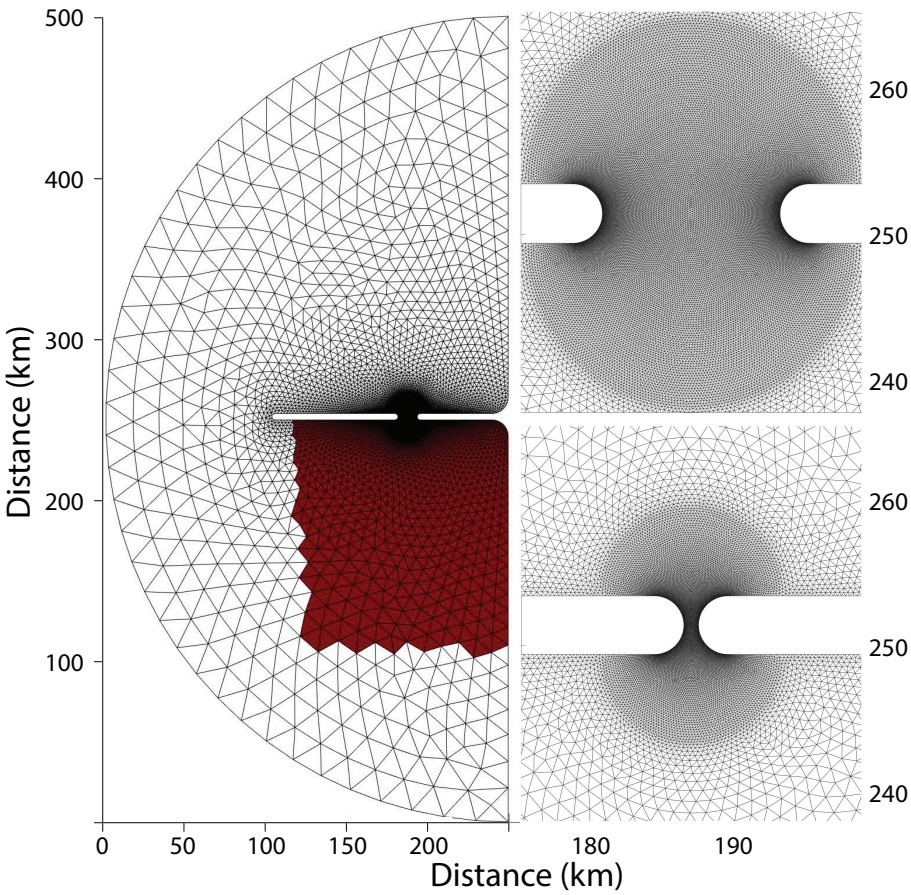

**Figure 1.** Left panel: The entire model domain with the peninsula attached to the eastern coast and the island located west of the peninsula. The red color marks the area with initial tracer concentration equal $1m^{-3}$. Right panel: The mesh near the strait with 12 km width (top) and 1 km width (bottom).

panel in Figure 1). The northern edge of the initial tracer release is at the center of the strait. This configuration of the initial concentration restricts the tracer exchange in the north-south direction to be through the strait only.

## 3  Overview of model results

By visual inspection, we observe that vortices form in all the different strait configurations. However, only a fraction of the straits produces self-propagating dipoles. Figure 3 provides an overview of all the simulations and we mark all straits where self-propagating dipoles are visually observed. The dipole formation clearly depends on the strait geometry, where narrow and short straits favors dipole formation. Additionally, with stronger tidal forcing ($A_t = 1.0$ m), dipoles form in wider and longer

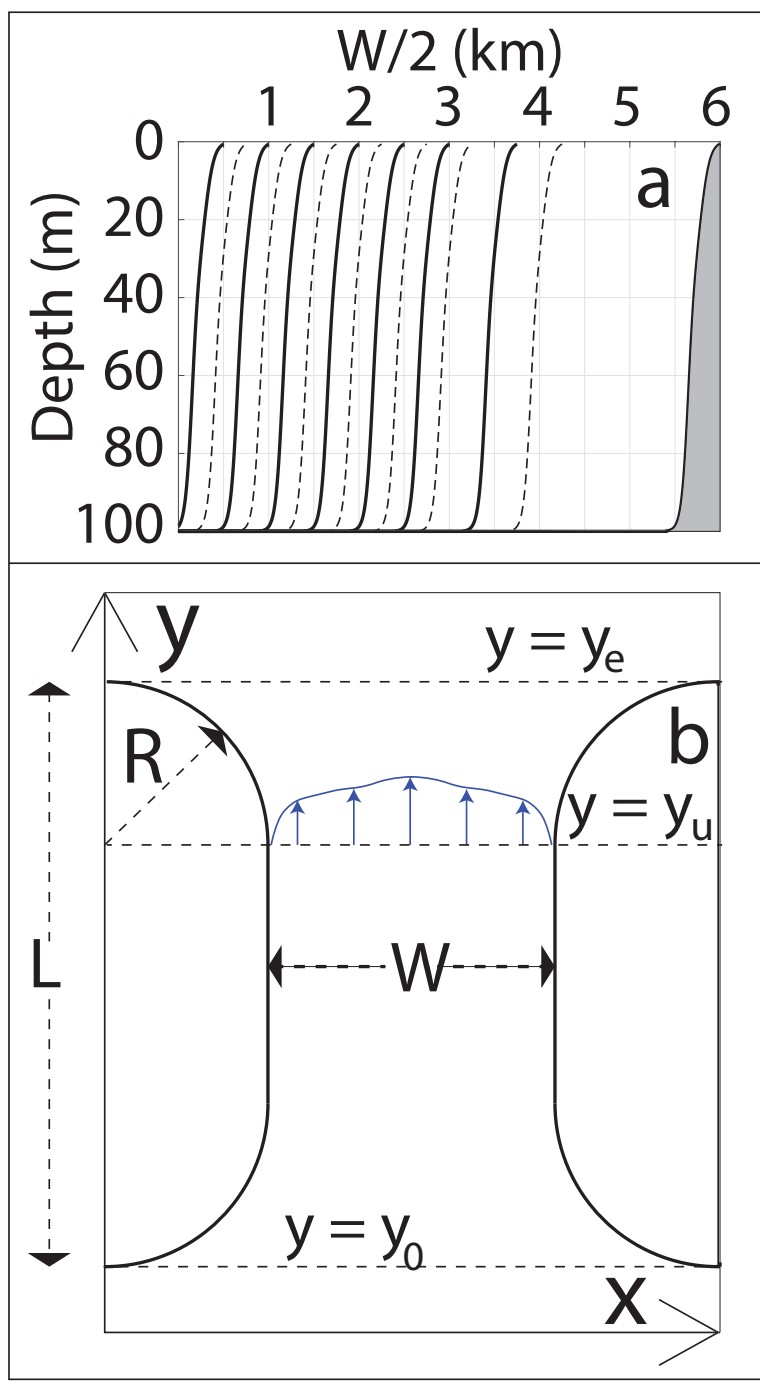

**Figure 2.** a) Vertical cross-section of bottom topography from the strait center to the eastern coastline for the different strait widths. The solid and dashed lines are used to more easily differentiate between the different strait widths. b) The coordinate system of the strait configuration.

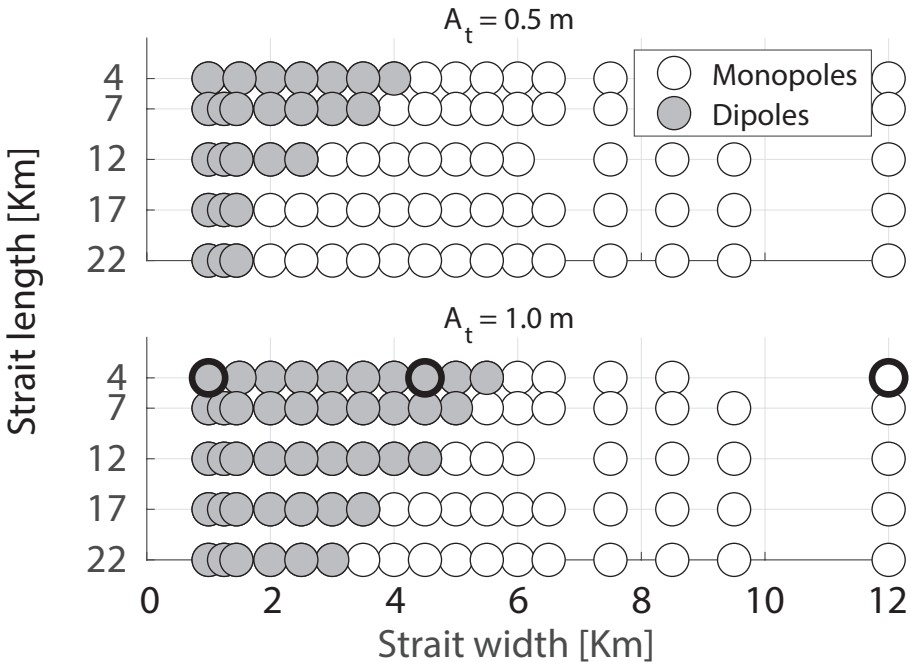

**Figure 3.** Overview of all simulations performed in this study. Gray color marks simulations where self propagating dipoles are formed. The upper and lower panel displays simulations forced with a tidal wave height amplitude of $A_t = 0.5$ m and $A_t = 1$ m, respectively. The three black thick circles mark the three simulations shown in Figures 4 to 6

.

straits compared to when the tidal forcing is weak ($A_t = 0.5$ m). In this section, we present an overview of the results illustrated by the temporal evolution of the tracer and vorticity distribution in three simulations.

We choose to show three examples where the tidal forcing and the strait length are similar ($A_t = 1$ m and $L = 4$ km), while the strait widths are $W = 1$ km, $W = 4.5$ km and $W = 12$ km respectively. The difference in strait width results in different temporal evolution of the tracer distribution and the vorticity fields. We show the results from the first half of the tidal cycle, where we define the tidal cycle to start at slack tide after ebb. The first six hours (t = 0-6 hours) are during flood tide and the tidal current is directed northward. All three examples have flow separation and vortex formation at the strait exit, but only in

the two former do the vortices connect into self-propagating dipoles.

    In the narrowest strait ($W = 1$ km, Figure 4), the flow separates 1.5 hours after slack tide. At this point the flow is dominated by two separated shear layers with negative (right) and positive (left) vorticity. The separated shear layers are connected by a trailing jet to the two initial vortices, which now form a self-propagating dipole. The dipole at this stage consists of two intense eddy cores filled with water having tracer concentration near 1. After 3 hours, the dipole has become larger and the vortex

cores are somewhat less intense. The outer part of the dipole now consists of water with near zero vorticity and near zero tracer concentration. The streamlines indicate that this low concentration water has not come through the strait but is entrained into





the dipole at the northern side of the strait. The dipole continues to grow while moving northward, fed by the trailing jet and by entrainment of low vorticity water. Since the dipole is formed early in the tidal cycle, the dipole has time to propagate far northward before the flow turns.

In the 4.5 km wide strait (Figure 5) the time period from slack tide till flow separation and dipole formation is longer than in the 1 km wide strait. At 1.5 hours, separation has not yet occurred. The vorticity is confined to the narrow viscous boundary layers, while the tracer has started to exit the strait. The width of the two boundary layers is similar to the 1 km strait. However, since the strait is wider, the boundary layers occupies a smaller fraction of the strait. Most of the water flowing through the strait therefore has near zero vorticity. At 3 hours, a dipole has formed and grows while moving northward during the tidal

period. The vorticity is mainly located inside the vortex cores and most of the dipole consist of water with near zero vorticity. An obvious difference from the 1 km wide strait (Figure 4) is that much of the near zero vorticity water in the dipole has come through the strait and contains tracer. This leads to a pattern where the tracer covers a larger area than the vorticity. The dipole barley detaches from the coastline before the flow reverses, and no proper trailing jet is formed. Instead, we observe a continuous vortex shedding from the separated shear layer at the strait exit, which to some degree interact and merge with the

stronger initial vortices.

In the widest strait ($W = 12$ km), we observe a continuous vortex shedding from the boundary layer similar to the 4.5 km wide strait (Figure 6). However, the vortices never interact across the width of the strait to form a dipole. In addition to a larger separation distance between the vortices at each side of the strait, the flow also separates later than in the two former examples. The vortices observed at the exit three hours after slack tide are advected through the strait and they are not formed

at the northern exit during the ongoing tidal phase. First, after almost four hours, are the first vortices shed from the separated boundary layer. These vortices do not interact across the strait to form a dipole, but rather seem to interact and merge with other co-rotating vortices at the same side of the strait. Since no self-propagating dipoles are formed, the vortices do not escape the return flow, and the net tracer transport through the strait is near zero.

The three examples shown in Figures 4 to 6 all have the same channel length, but they illustrate the process of dipole for-

mation and dipole transport properties. These processes are similar for all channel lengths, although channel length influences channel flow and whether dipoles are formed or not. In general, longer straits require narrower strait widths for dipoles to form (Figure 3), and flow separation and vortex formation occurs later in the tidal cycle.

In the following, we go into the details of flow separation, vortex formation and dipole properties. These topics are important for the understanding of how strait geometry affects flow dynamics and water exchange through narrow tidal straits.

## 4   Flow separation and vortex formation

The timing of flow separation depends on the flow dynamics at the strait exit. Here the balance between non-linear advection and pressure forces leads to an adverse pressure gradient caused by the widening of the strait. The flow separates from the coastline when the adverse pressure gradient acts in the same direction as the friction and brings the velocity in the viscous boundary layer to zero (Signell and Geyer, 1991; Kundu, 1990). Since the adverse pressure gradient results from the nonlinear




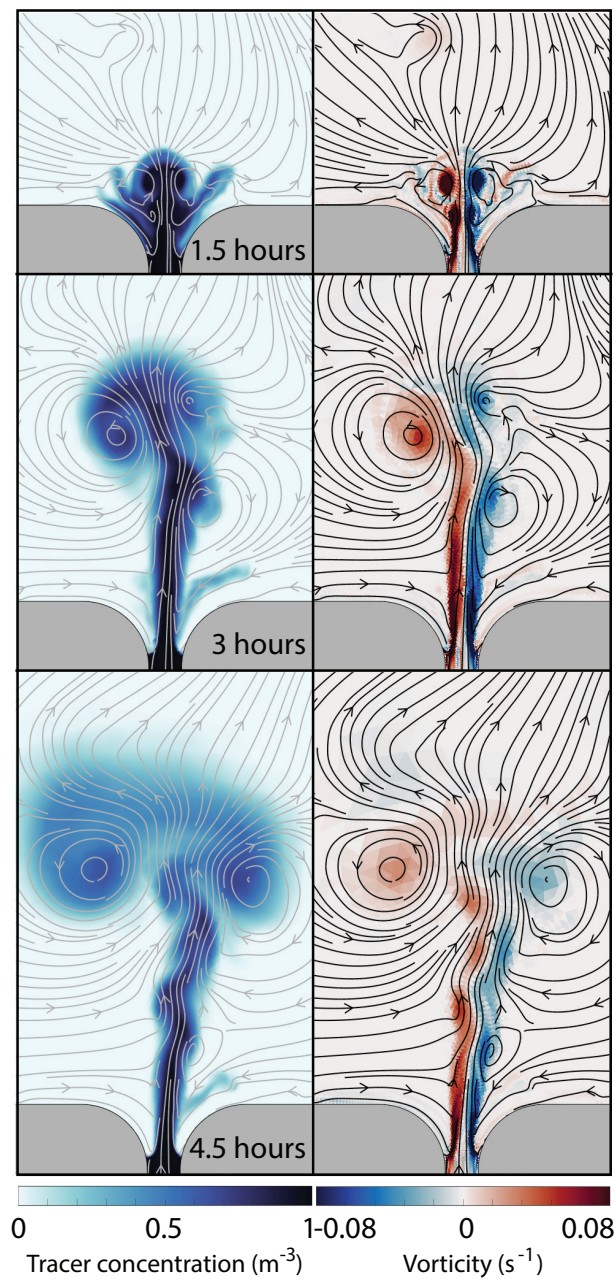

0          0.5          1 -0.08     0     0.08
Tracer concentration (m$^{-3}$)          Vorticity (s$^{-1}$)

**Figure 4.** The temporal tracer and vorticity fields, with the corresponding stream-function, is displayed for a 1 km wide and 4 km long strait in the left and right panel, respectively. The experiment is forced with a tidal wave of amplitude $A_t = 1$ m. The upper, middle and lower panels shows a snapshot in time of the tracer and the vorticity fields at 1.5 hours, 3 hours and 4.5 hours after slack tide, respectively.

advection, the separation time can be related to the ratio of the local acceleration to the nonlinear advection, also called the





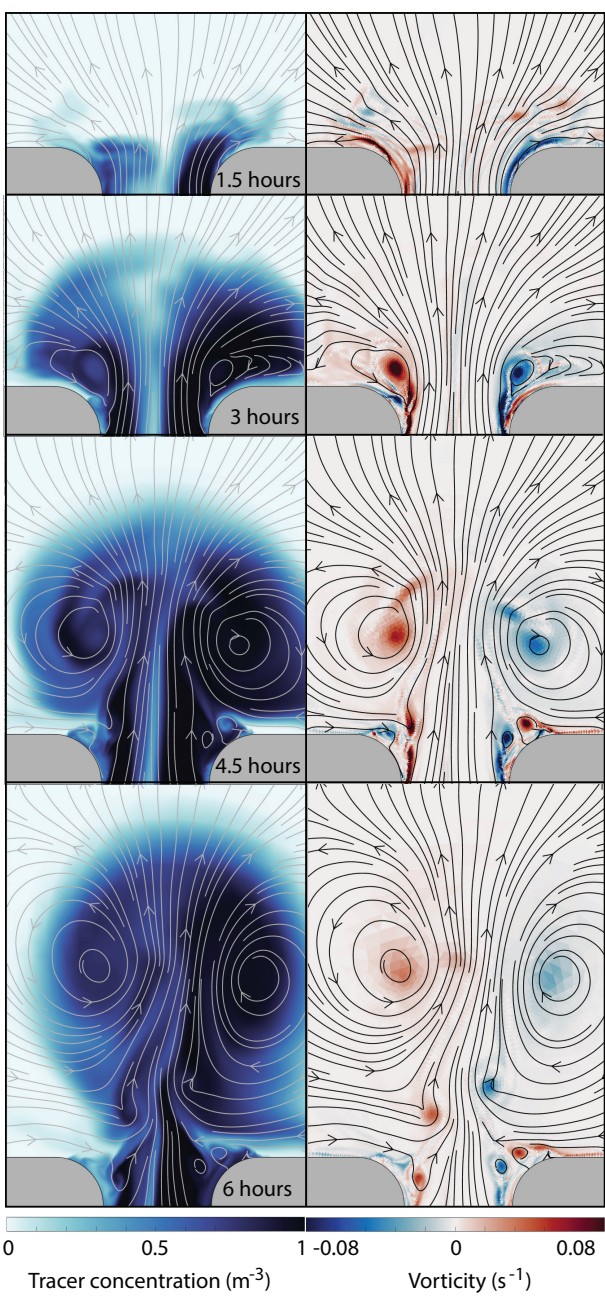

**Figure 5.** The temporal tracer and vorticity fields, with the corresponding stream-function, is displayed for a 4.5 km wide and 4 km long strait in the left and right panel, respectively. The experiment is forced with a tidal wave of amplitude $A_t = 1$ m. The upper, middle and lower panels shows a snapshot in time of the tracer and the vorticity fields at 1.5 hours, 3 hours and 4.5 hours after slack tide, respectively.

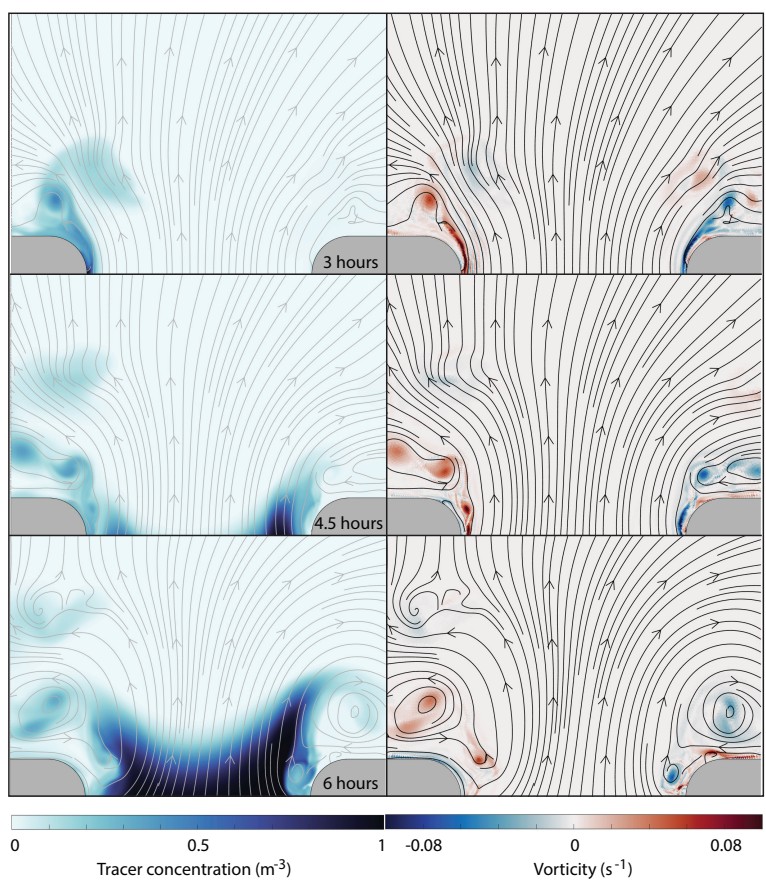

0 0.5 1 -0.08 0 0.08
Tracer concentration (m⁻³) Vorticity (s⁻¹)

**Figure 6.** The temporal tracer and vorticity fields, with the corresponding stream-function, is displayed for a 12 km wide and 4 km long strait in the left and right panel, respectively. The experiment is forced with a tidal wave of amplitude $A_t = 1$ m. The upper, middle and lower panels shows a snapshot in time of the tracer and the vorticity fields at 1.5 hours, 3 hours and 4.5 hours after slack tide, respectively.

Keulegan-Carpenter number ($K_c$) (Signell and Geyer, 1991). Flow separation can occur when

$$K_c = \frac{UT^*}{R} > 1,  \tag{10}$$

where $T^*$ is the timescale where the flow dynamics becomes non-linear and $R$ is the length scale of the strait exit (Figure 2b). From here and through the rest of the paper, the velocity scale $U$ is given by the tidal velocity amplitude. This is calculated as the maximum in time of the cross-strait average at $y = y_u$ (see Figure 2 for coordinate definitions). Assuming the time of separation, $T_s$, can be related to $T^*$ and that $K_c$ must obtain a certain value for separation to occur, then $T_s$ should be proportional to $R/U$. This relation is confirmed when plotting $T_s$ against $R/U$ (see Figure 7). Here, $T_s$ is the separation time obtained from the model results (details of how this is done are given below). Corresponding values of $K_c$ lays mainly between 5 and 15.



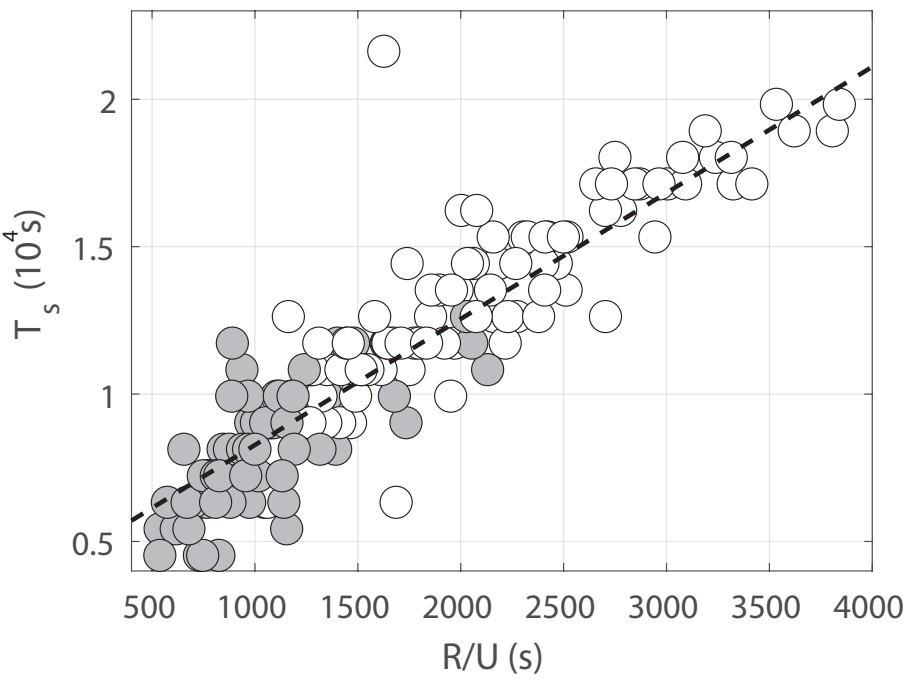

**Figure 7.** Separation-time ($T_s$) plotted against ($R/U$). The dashed line is the best linear fit $T_s = 4.3R/U + 3998$. Straits with self propagating dipoles are marked gray.

The formation of starting vortices and self-propagating dipoles occurs when the flow separates. The vorticity needed to form these vortices results from the strong velocity front that is formed at the boundary between the newly separated flow and the reversed flow along the coast. At the time of flow separation, the velocity front immediately rolls-up into a vortex. This process is illustrated in Figure 8, where the flow field near the point of separation is plotted on top of vorticity and surface elevation for the same three simulations shown in Figures 4 to 6. The vorticity created in the velocity front causes a maximum absolute

value of vorticity to occur at separation time. This is shown in Figure 9 for the same three simulations shown in Figure 8.

         For the simulations with strait widths 1 km and 4.5 km (upper and middle panel of Figure 8) the two initial vortices interact and form a dipole. In these two straits, we see a rapid buildup towards the maximum absolute value in vorticity followed by a decrease (black and green curve in Figure 9). In the 1 km wide strait, the initial vortices remain attached to the strait by a trailing jet, and we observe only one prominent peak in maximum absolute value of vorticity (black curve in Figure 9). In

the 4.5 km wide strait, several vortices are shed from the separated velocity front after the initial vortex shedding (see Figure 5), and several local maximums are occurring after flow separation (green curve in Figure 9). In the widest strait, the initial vortices never connect into a dipole, and the maximum absolute value of vorticity is much less prominent compared to the narrower straits (blue curve in Figure 9). However, also for the widest strait, we observe by visual inspection that the maximum absolute value of vorticity coincides with the initial vortex formation due to flow separation, at about 4 hours after slack tide.

We find that, for all simulations, the maximum absolute value of vorticity corresponds to the separation time. Therefore, the


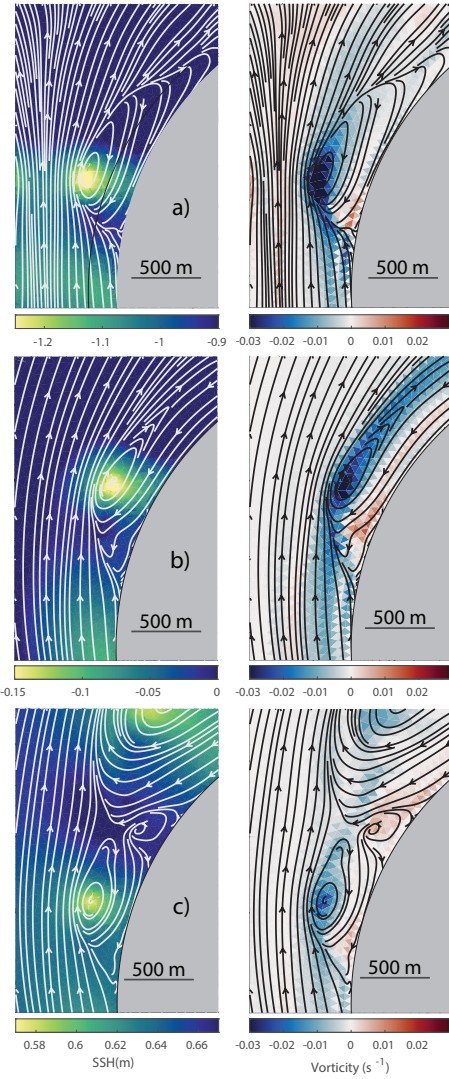

**Figure 8.** Sea surface height (left) and the vorticity (right), with contours showing the corresponding stream lines, are shown at separation time. We show the fields in the three straits displayed in a) Figure 4 ($W = 1$ km), b) Figure 5 ($W = 4.5$ km), and c) Figure 6 ($W = 12$ km), respectively.

separation time is estimated from the timing of the absolute value of vorticity within the strait exit ($y_u < y \leq y_e$, see Figure 2b).

The initial vorticity of the vortices created during flow separation is an important parameter, determining whether the vortices formed on each side of the strait are connected to form a dipole or not, as well as the propagation velocity of the dipole that 230  forms. Here, the vortices are represented by the radial profiles of Lamb-Oseen (LO) vortices (Lamb, 1916; Leweke et al.,


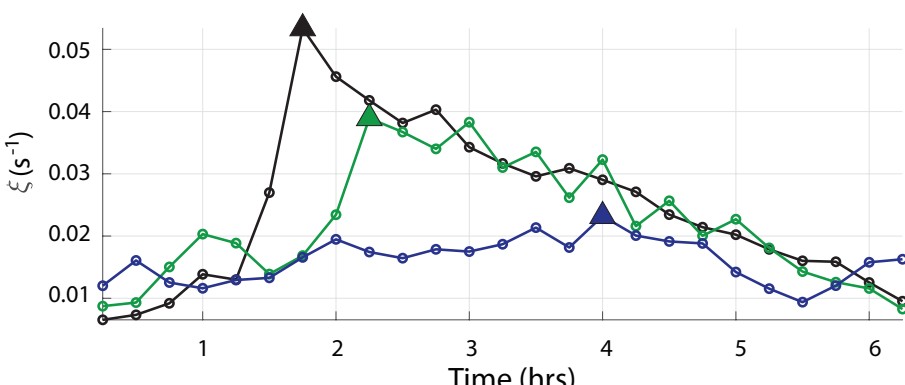

**Figure 9.** Time-series of the maximum magnitude of vorticity at the strait exit, defined as the area where $y_u < y \leq y_e$. The black, green and blue curves represents the same three simulations shown in Figure 4 to 6, respectively. The triangles mark the separation time.

2016),

$$v_\theta = \frac{\Gamma}{2\pi r}(1 - e^{-\frac{r^2}{a^2}}) \tag{11a}$$

$$\xi = \frac{\Gamma}{\pi a^2}e^{-\frac{r^2}{a^2}}, \tag{11b}$$

where $\xi$ is the vorticity, $\Gamma$ is the circulation of the vortex, $a$ is the radius of the vortex core and $r$ is the distance from the center

of the vortex core. Originally $a$ is increasing with time and depends on viscosity. Equation 11 is a particular solution to the Navier-Stokes equations (Habibah et al., 2018), and is known to show good agreement with experimental data (Leweke et al., 2016). The vortex shape described by Eq. 11 fits well to our modelled vorticity (see Figure 10). We obtain the core radius, $a$, by finding the best fit of Eq. 11 to the modelled vortices, using the maximum and minimum vorticity from the model data.

From the results shown in Figure 8, we see that the newly formed vortices have nearly equal size, even though the three

simulations have very different characteristics. The estimation of core radius for all 164 simulations shows that what is indicated by Figure 8 is a general result. The estimated core radius at separation time is given by $a(T_s) = 110 \pm 18$ m for all simulations and $a(T_s) = 116 \pm 14$ m for the dipoles (mean $\pm$ one standard deviation, see Figure 11). This suggests that the vortex core radius is near constant across all simulations, which again suggests that the vorticity should be proportional to the strait velocity. Plotting the maximum absolute value of the vorticity against the along-strait velocity at separation time $v(T_s)$ (Figure

12), suggests that the maximum absolute value of vorticity can be represented as

$$|\xi(T_s)|_{max} \simeq \frac{|v(T_s)|}{a(T_s)}. \tag{12}$$

We have shown that the flow separation coincides with a maximum in absolute value of vorticity and that the dipole is formed at the time of separation. The vorticity of the initial vortices are given by strait velocity divided by the core radius, and the initial core radius is near equal for all simulations. In the following section, we describe how dipole vortices are recognized

and the determination of their propagation velocity.



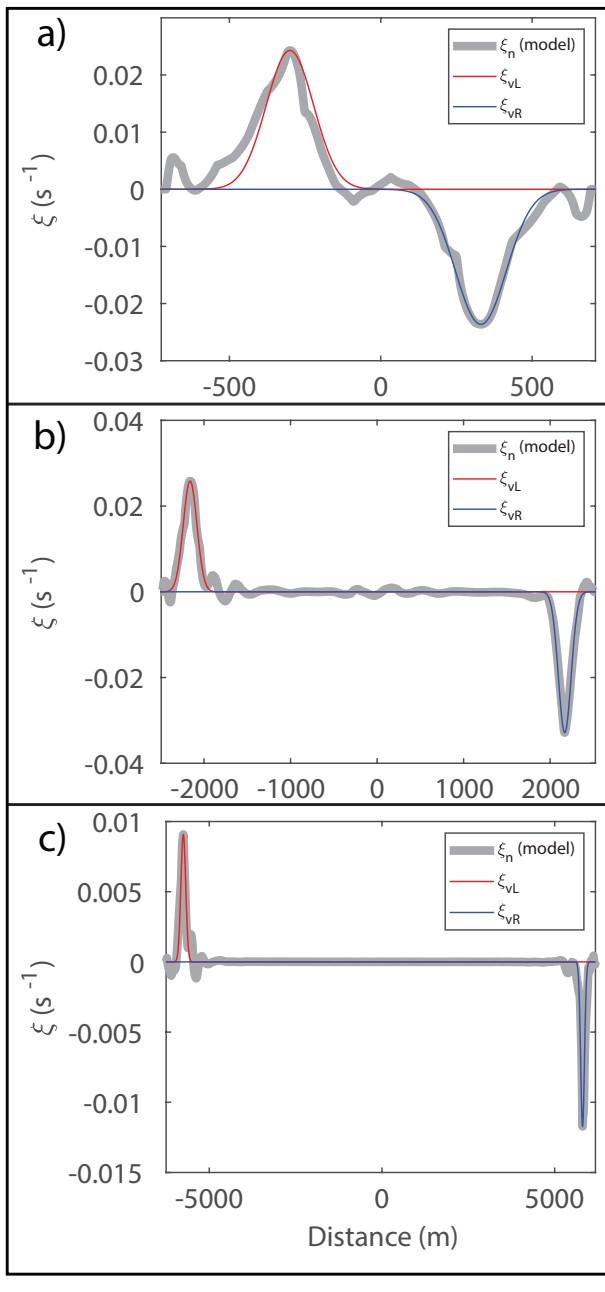

**Figure 10.** The vorticity distribution along a line intersecting the two vortices at each side of the strait at separation time. a), b) and c) is from the three simulations shown in Figures 4 to 6, respectively. The gray line shows the vorticity distribution from the model output, while the red and blue lines are calculated vorticity distribution using Eq. 11b, for the left and right vortex, respectively.



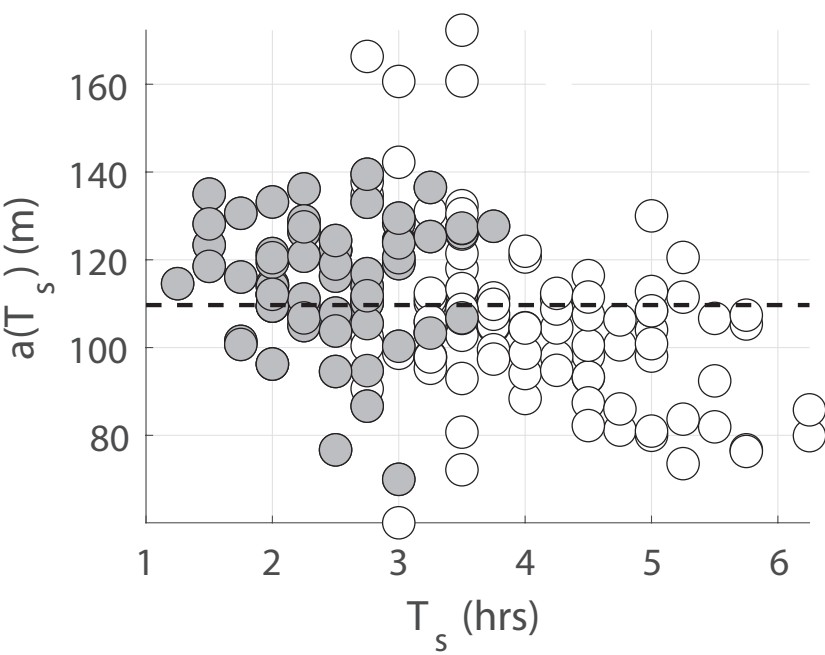

**Figure 11.** The vortex core radius at separation time, plotted against the separation time. The core radius is the mean radius of the two vortex cores formed at each side of the strait. Straits with self propagating dipoles are marked gray.

## 5 Dipole recognition and tracking

To obtain the dipole properties we track the initial vortices from the time of flow separation to the end of the tidal phase. The vortices are points of minimum surface elevation as seen in Figure 8. So, when tracking the vortices, we simply track the minima in surface elevation. Typically, vortices form simultaneously on each side of the strait at separation time, and we start

tracking the minimum surface elevation on each side of the strait from this moment. We evaluate the propagation velocity and direction of the two vortices to determine whether they have connected into a dipole or not, using two criteria, illustrated in Figure 13.

The criteria are based on two simple principles. The first criterion is that a dipole will propagate normal to the line connecting two vortices and therefore conserve the distance between them (Leweke et al., 2016). We observe that vortices that do not

connect into dipoles tend to be advected to each side of the strait, increasing the distance between them. The second criterion is based on the fact that a dipole escaping the returning tidal flow needs to have a propagating velocity over a certain limit. Fitting these two criteria to the results of visual inspection leads to the following formulations, which are used to recognize dipoles in the simulation results (see Figure 13 for notations),

$$\frac{2(y_2 - y_1)}{b_2 - b_1} < 2.9, \tag{13}$$





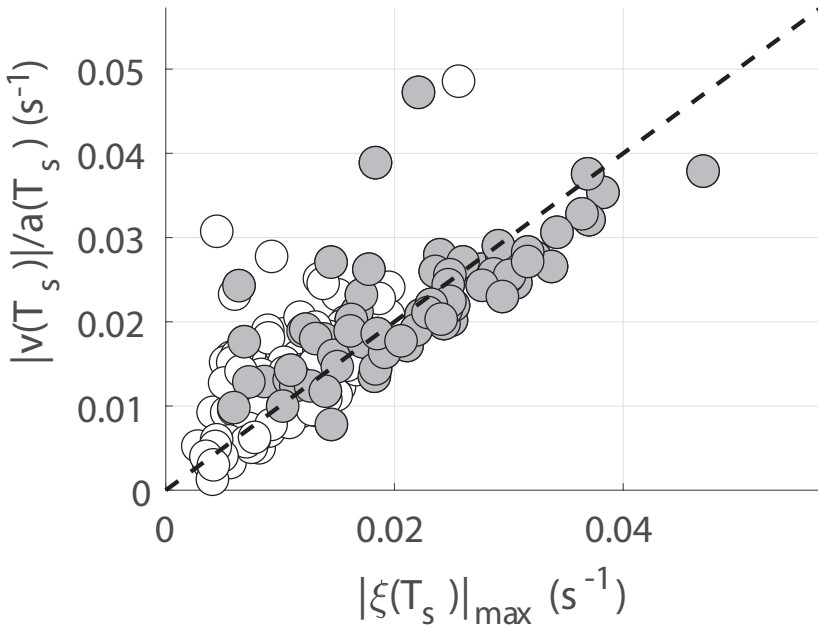

**Figure 12.** The theoretical velocity shear $v/a$ plotted against the absolute value of the vorticity in the vortices at separation time. Straits with self propagating dipoles are marked gray.

and

$$U_{dip} = \frac{y_2 - y_1}{\Delta t} > 0.2 ms^{-1}. \tag{14}$$

The first of these criteria sets a limit to the increase in distance between the vortices compared to northward propagation of the dipole, while the second criterion requires that the dipole have a mean propagation speed larger than 0.2 m/s. $\Delta t$ is the time between the two dipole positions given by $y_1$ and $y_2$. The last criterion is important to rule out dipoles that form late in the 270 tidal cycle and will not escape the strait before the tidal current reverses. These dipoles often move slowly and do not move out of the strait, and due to this their separation distance is near constant because it is restricted by the coastline. To recognise escaping dipoles, we find that it is necessary to set a lower limit to their propagation velocity and therefore we have introduced the second criterion defined by Eq. 14.

When tracking the vortices, we obtain the dipole propagation velocities, which together with the velocity and vorticity 275 distributions, enables us to investigate the vortex properties.



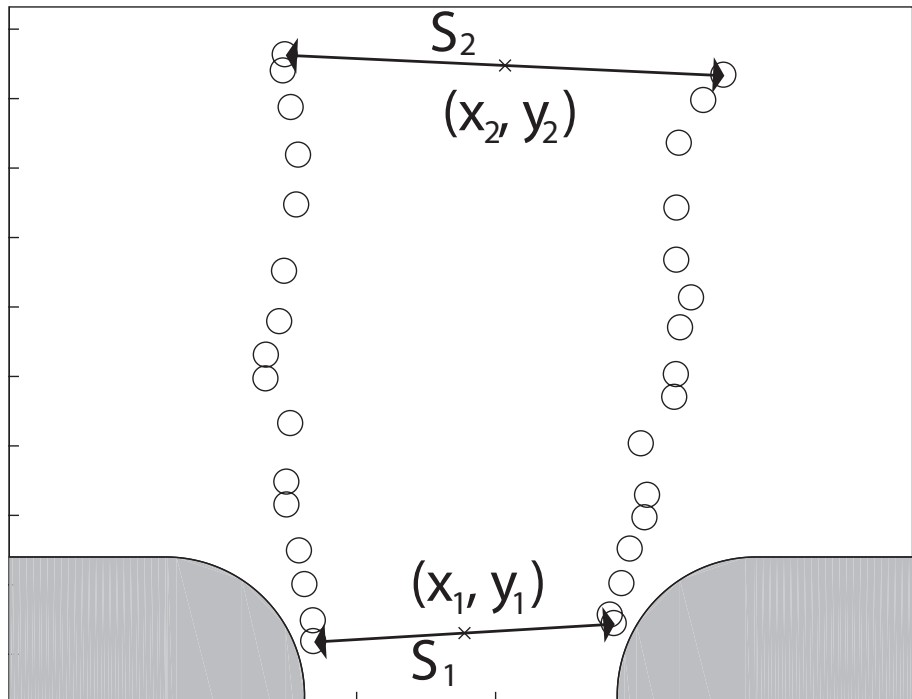

**Figure 13.** A sketch illustrating the dipole tracking. $x_1$ and $y_1$ is the position of the midpoint between the two vortices, and b1 is the distance between the two vortices at separation time, $t = T_s$. Likewise, $x_2$, $y_2$, is the position on the midpoint between the vortices, and $b_2$ is the distance between the vortices at $t = T_s + \Delta t$.

## 6   Representation of the dipole propagation velocity

Dipole properties, such as core radius ($a$) and propagation velocity ($U_{dip}$) determines the net water exchange through the strait (Kashiwai, 1984a; Wells and van Heijst, 2003). Another important parameter is the sink radius ($R_s$). The water volume within the half circle with radius $R_s$ will be drawn into the strait when the flow reverses at $t = T/2$. If the dipole has travelled a

distance larger than $R_s$, it will escape the return flow. Here, we choose to investigate dipole properties inside the sink radius.

Comparing the tracked dipole velocities to the theoretical velocities obtained from Eq. 1, we find that the dipole propagation velocity given by Eq. 1 is too low. Instead, we get a much better fit when using the sum of the contributions from the two vortices,

$$U_{dip} \simeq \frac{|\Gamma_1| + |\Gamma_2|}{2\pi b}, \tag{15}$$

where $\Gamma_1$ and $\Gamma_2$ are the circulation of the two vortices respectively. We calculate $\Gamma_1$ and $\Gamma_2$ from Eq. 11 using the value of maximum vorticity

$$\Gamma = \pi a^2 \xi_{max}, \tag{16}$$





and compare the dipole propagation velocity estimated using Eq. 15 to the tracked velocities. Figure 14 shows the comparison for each time-step in the same two simulations shown in Figures 4 and 5, and Figure 15a shows the comparison for dipole velocities averaged within the sink radius.

Assuming the two vortices are equal gives

$$U_{dip} \simeq \frac{\Gamma}{\pi b}. \tag{17}$$

Since the majority of the vorticity is contained within the core radius, scaling analysis suggests the circulation $\Gamma \simeq \pi a U$, which is obtained by assuming $\xi \simeq U/a$ . This suggests that the dipole propagation velocity can be represented as

$$U_{dip} \simeq \alpha U, \tag{18}$$

where $\alpha = a/b$ is the aspect ratio of the vortices. The comparison to tracked velocities (Figure 15b) shows that Eq. 18 is a good representation of the dipole propagation velocity.

The dipole propagation velocity is crucial when determining the transport properties of the dipole in relation to tidal pumping (Kashiwai, 1984a; Wells and van Heijst, 2003). In the next section we will use the simple relations found here in the search for a parameter describing the net water exchange through the strait.

## 7 Water exchange through the strait

### 7.1 Effective tracer transport

To investigate the role of dipole vortices in setting the net water exchange, we first quantify the effective tracer transport $Q_e$,

$$Q_e = \frac{Q_n}{Q_m}. \tag{19}$$

$Q_e$ is the ratio between the net tracer transport, $Q_n$, and the maximum potential for net tracer transport through the strait, $Q_m$, over the course of one tidal cycle. $Q_n$ is calculated through a cross-section in the center of the strait ($y = y_e - L/2$) as

$$Q_n = \sum_{t=0}^{T} \sum_{n=1}^{N} c_n v_n dA_n dt. \tag{20}$$

Here $v_n$ is the normal velocity through an area element $dA_n$, and $c_n$ is the tracer concentration in grid cell $n$. $Q_m$ is given by

$$Q_m = \sum_{t=0}^{T/2} \sum_{n=1}^{N} c_{max} v_n dA_n dt + \sum_{t=T/2}^{T} \sum_{n=1}^{N} c_{min} v_n dA_n dt. \tag{21}$$

The maximum possible tracer transport occurs when the northward transport consists entirely of water containing tracer concentration $c = c_{max}$, and the southward transport consists entirely of water containing tracer concentration $c = c_{min}$. In our case $c_{max} = 1 \, m^{-3}$ and $c_{min} = 0 \, m^{-3}$. The effective tracer transport is independent of the volume transport, and is a measure of how efficient water is exchanged through the strait.



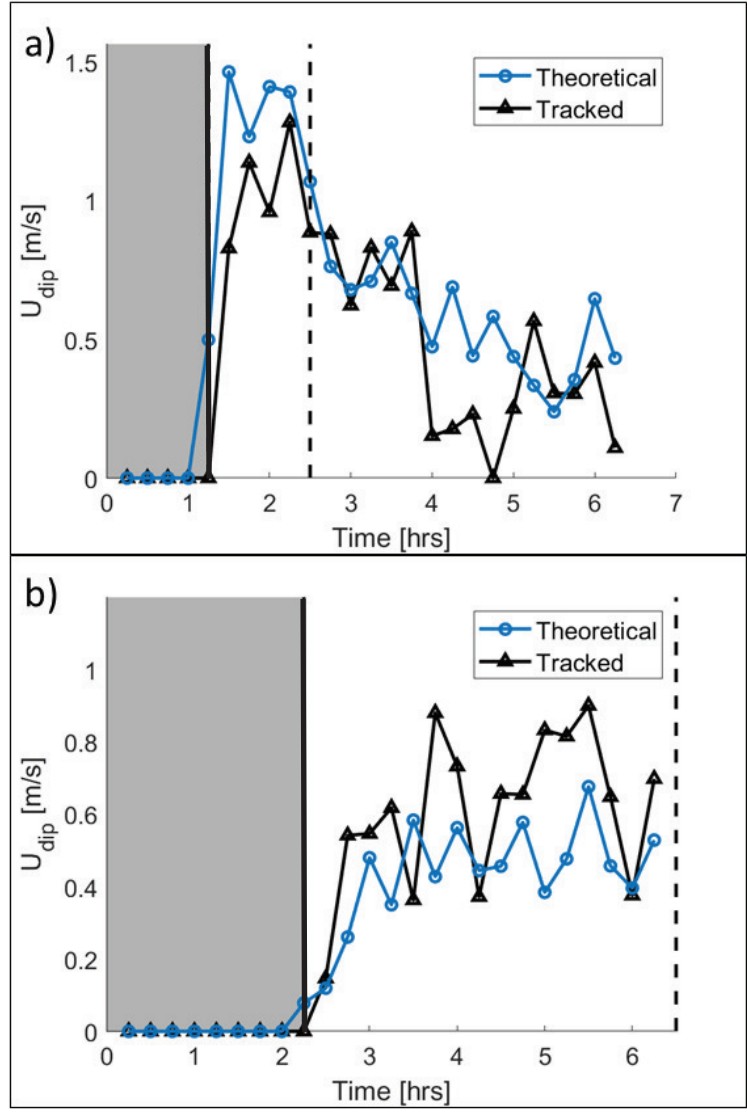

**Figure 14.** Dipole propagation velocity for a dipole formed in (a) the 1 km wide and 4 km long strait shown in Figure 4, and (b) the 4.5 km wide and 4 km long strait shown in Figure 5. The black curves are velocities obtained from dipole tracking, while the blue curves are velocities calculated using Eq. 15. The gray patch indicates the time before flow separation. The dashed black line indicates when the dipole escapes the sink region.

## 7.2 Water exchange by self-propagating dipoles

How effective the dipole vortices are in exchanging water through a strait depends on whether the dipole escapes the return flow or not (Kashiwai, 1984a; Wells and van Heijst, 2003). Both Kashiwai (1984a) and Wells and van Heijst (2003) investigated



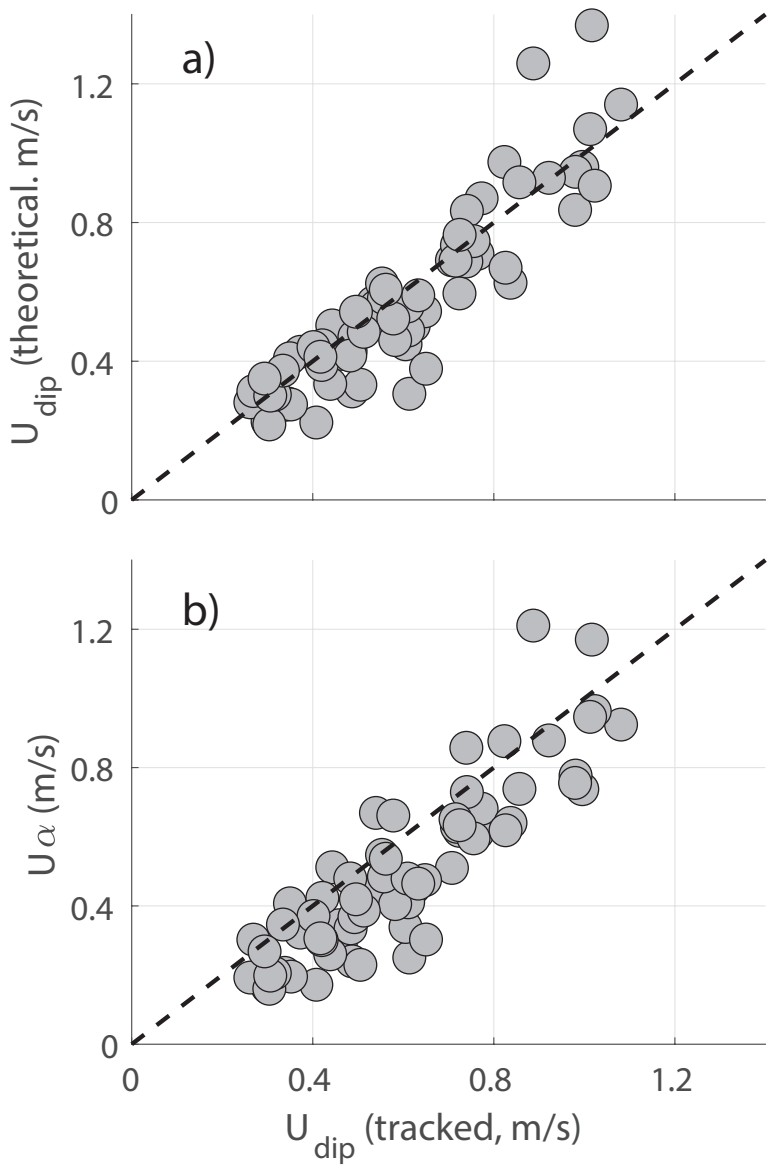

**Figure 15.** The dipole velocities obtained from tracking (on the x-axis) plotted against a) the theoretical dipole velocities (Eq. 15), and against $U\alpha$ (Eq. 18) in the lower panel. The tracked velocities, theoretical velocities and $\alpha$ are averaged over the time period when the dipole is located inside the sink region.

the dipole position relative to the sink radius when the flow reverses to evaluate the potential for the dipole to escape. While Kashiwai (1984a) only considered the position of the dipole relative to the sink region, Wells and van Heijst (2003) evaluated the magnitude of the return flow relative to the dipole velocity at its position. Both approaches resulted in a threshold value





of the Strouhal number ($St_c$) between 0.8 and 0.13, separating the dipoles escaping ($St < St_c$) and dipoles not escaping ($St > St_c$) the return flow.

We follow the approach of Kashiwai (1984a) and investigate the dipole transport potential by evaluating the length of the dipole propagation distance, $L_d$, relative to the sink radius, $R_s$, at $t = T/2$.

$$L_d = U_{dip} \left( \frac{T}{2} - T_s \right), \tag{22}$$

and $R_s$ is given by

$$R_s = \sqrt{\frac{2Q}{\pi H}} = \frac{\sqrt{2WUT}}{\pi^2}, \tag{23}$$

where $W$ is channel width, $Q \simeq WH \int_0^{T/2} v dt = WHUT/\pi$ is the tidal prism, and $v = U sin(\omega t)$ is the along-strait velocity. Here we assume the sink region is formed as a half circle, with a radius, $R_s$, and the water depth, $H$, is constant inside the domain.

The position of the dipole relative to the sink radius at $t = T/2$ is evaluated by the non-dimensional parameter $S_d$,

$$S_d = \frac{R_s}{L_d} = \frac{\sqrt{2WUT}}{\pi U_{dip} \left( \frac{T}{2} - T_s \right)}. \tag{24}$$

This expression is formulated in the same fashion as the Strouhal number by Kashiwai (1984a) and Wells and van Heijst (2003), meaning that low numbers favor escaping dipoles and effective water exchange. If $S_d > 1$ the dipole is inside the sink region when the flow reverses, and conversely, if $S_d < 1$ the dipole is outside the sink region and will escape the return flow.

$S_d$ considers dipole transport properties only, and shows different behavior for the different strait lengths, when plotted against effective tracer transport (Figure 16a). Values of $S_d$ well below one does not guarantee net tracer transports, as can be seen for some of the longest straits shown in Figure 16a. This indicates that we need to consider strait length in order to describe the effective tracer transport through the strait.

The dipole can only be an important contributor for water exchange if the strait is shorter than the tidal excursion. If the

strait is longer than a tidal excursion, the water mass on one side of the strait will not be able to travel through the strait, with zero net tracer exchange as a result. In order to evaluate the effect of strait length we introduce the nondimensional length scale $S_L$,

$$S_L = \frac{L}{L_t} = \frac{\pi L}{UT}. \tag{25}$$

Here, $L_t = \int_0^{T/2} v dt$ is the tidal excursion and $L$ is the strait length. If $S_L > 1$, the tracer front will not propagate through the

strait during one half tidal cycle and no tracer will be available for the dipole to capture and transport away from the strait. This is the case for many of the long straits, with zero tracer transport as a result (Figure 16b). However, similar as for $S_d$, $S_L < 1$ does not guarantee a net tracer transport.

$S_L$ and $S_d$ can be combined to give the effective tracer transport through the strait. To show this, we consider the situation where $S_L < 1$, which assures that tracer will flow through the channel. We apply a simple kinematic model illustrated by Figure

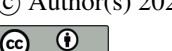


**Figure 16.** The effective transport, $Q_e$ plotted against the non-dimensional parameters a) $S_d$ and b) $S_L$.

17. This figure illustrates the tracer distribution at $t = T/2$, where the dark gray illustrates the tracer in the dipole, the medium gray illustrates the tracer in the jet following the dipole and the light gray is the tracer inside the channel. All the tracer inside the channel and an unknown fraction of the tracer in the jet and dipole will be drawn back into the channel when the flow turns at $t = T/2$. We assume that the fraction inside the sink region will be drawn back, but this fraction depends on the shape of the




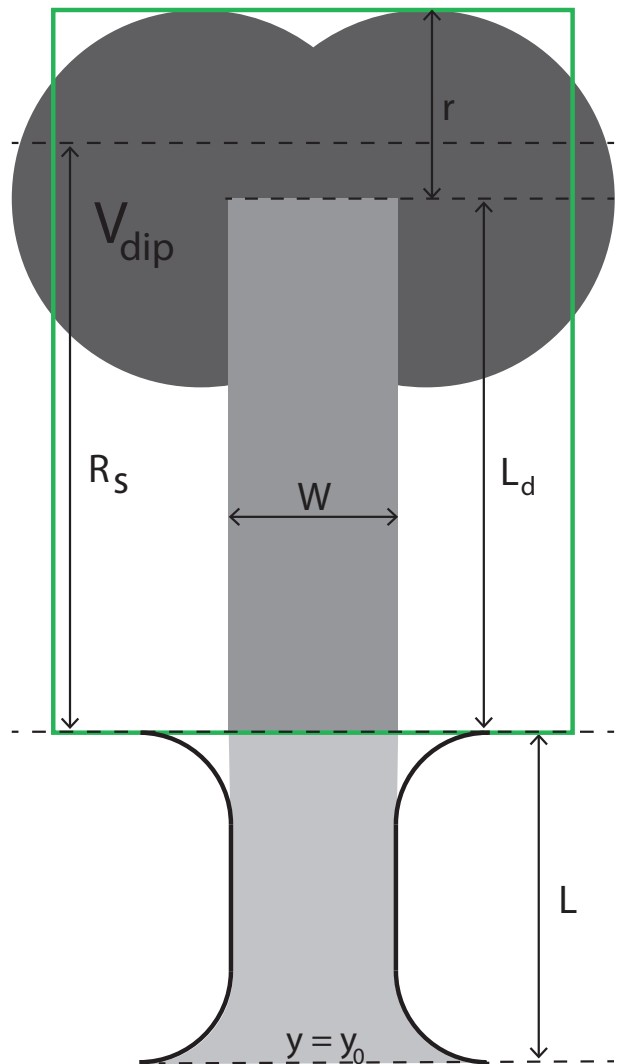

**Figure 17.** Idealized distribution of tracer at $t = T/2$ between the dipole (dark gray), jet (medium gray) and strait (light gray).

dipole and jet which is not easily estimated. However, to simplify the problem we assume that the dipole/jet is shaped like a

rectangle, as illustrated in green in Figure 17. The fraction inside $R_s$ is now given by the lengths $L_d$, $R_s$ and $r$ only. We have introduced the distance $r$ to include that some of the dipole can escape even if $L_d < R_s$.

At $t = 0$ we assume that the tracer front is located on one side of the strait at $y = y_0$, and that the water transported into the strait at $y = y_0$ always has a tracer concentration equal to $c_{max}$. The tracer transported through the cross-section at $y = y_0$ between $t = 0$ and $t = T/2$ is given by $c_{max} W H L_t$. The tracer distribution at $t = T/2$ is divided between the strait, jet and

dipole as illustrated in Figure 17. This can be expressed as

$$WL_t = WL + WL_d + V_{dip}, \tag{26}$$





where $V_{dip}$ represents the volume with tracer concentrations equal $c_{max}$ in the dipole. $H$ and $c_{max}$ cancels as they appear on both sides of the equation. If the water that is drawn back maintains its tracer concentration $c_{max}$ and the water that originates on the other side of the strait has a tracer concentration of $c_{min}$, the net tracer transport can be expressed as

$$q_n = \big(W(L_t - L) - \frac{R_s}{L_d + r}(W L_d + V_{dip})\big)c_{max}$$
$$- \frac{L_d + r - R_s}{L_d + r}(W L_d + V_{dip})c_{min}. \qquad (27)$$

Here, we assume that the net volume flux during one tidal cycle is zero. Combining Eqs. 26 and 27 gives

$$q_n = W(L_t - L)(1 - \frac{R_s}{L_d + r})(c_{max} - c_{min}). \qquad (28)$$

The maximum potential for tracer transport (see Eq. 21) is

$$q_m = W L_t(c_{max} - c_{min}). \qquad (29)$$

Dividing Eq. 28 by $q_m$ gives the effective tracer transport

$$q_e = (1 - S_L)(1 - \frac{R_s}{L_d + r}) \qquad (30)$$

Doing a series expansion of $(1 + r/L_d)^{-1}$ and ignoring terms of second order and higher gives

$$q_e = (1 - S_L)\big(1 - S_d(1 - \frac{r}{L_d})\big). \qquad (31)$$

The series expansion and the ignoring of higher order terms may not be strictly mathematically correct, because $r/L_d$ might be too large. However, what is important here is to reduce the effect of $S_d$, which is the effect in the original Eq. 30. Thus, using the simple kinematic model illustrated by Figure 17, we can express the effective tracer transport in a simple combination of $S_d$ and $S_L$, and the new parameter $r/L_d$. The result is shown in Figure 18, where we plot $q_e$ against $Q_e$ for two different values of $r/L_d$. $r/L_d$ works as a weighting of $S_d$, but even for $r/L_d = 0$ (Figure 18a) it is clear that the kinematic model capture the main physics of the problem. However, for $r/L_d = 0$, $q_e = 0$ for $S_d = 1$, which surely does not agree with the simulations (Figure 16a). With $r/L_d = 0.4$, $Q_e$ will be zero for $S_d(1 - r/L_d) > 1$. This can be seen from Figure 16a, where all transports for $S_d > 1.6$ is near zero. Making the same plot against $S_d(1 - r/L_d)$ will shift all transports above zero to $S_d(1 - r/L_d) < 1$. As can be seen from Figure 18b, $r/L_d = 0.4$ also collapses $Q_e$ onto the line given by $Q_e = q_e$. All results from simulations where dipoles is identified is located on or close to the line given by $Q_e = q_e$.

## 8 Discussion

### 8.1 Effect of strait length on flow dynamics

To understand why long straits produce less self-propagating dipoles than longer straits (Figure 3) it is instructive to use the model of Garrett and Cummins (2005). They consider the cross-strait mean velocity, $v$, which is a function of time and the



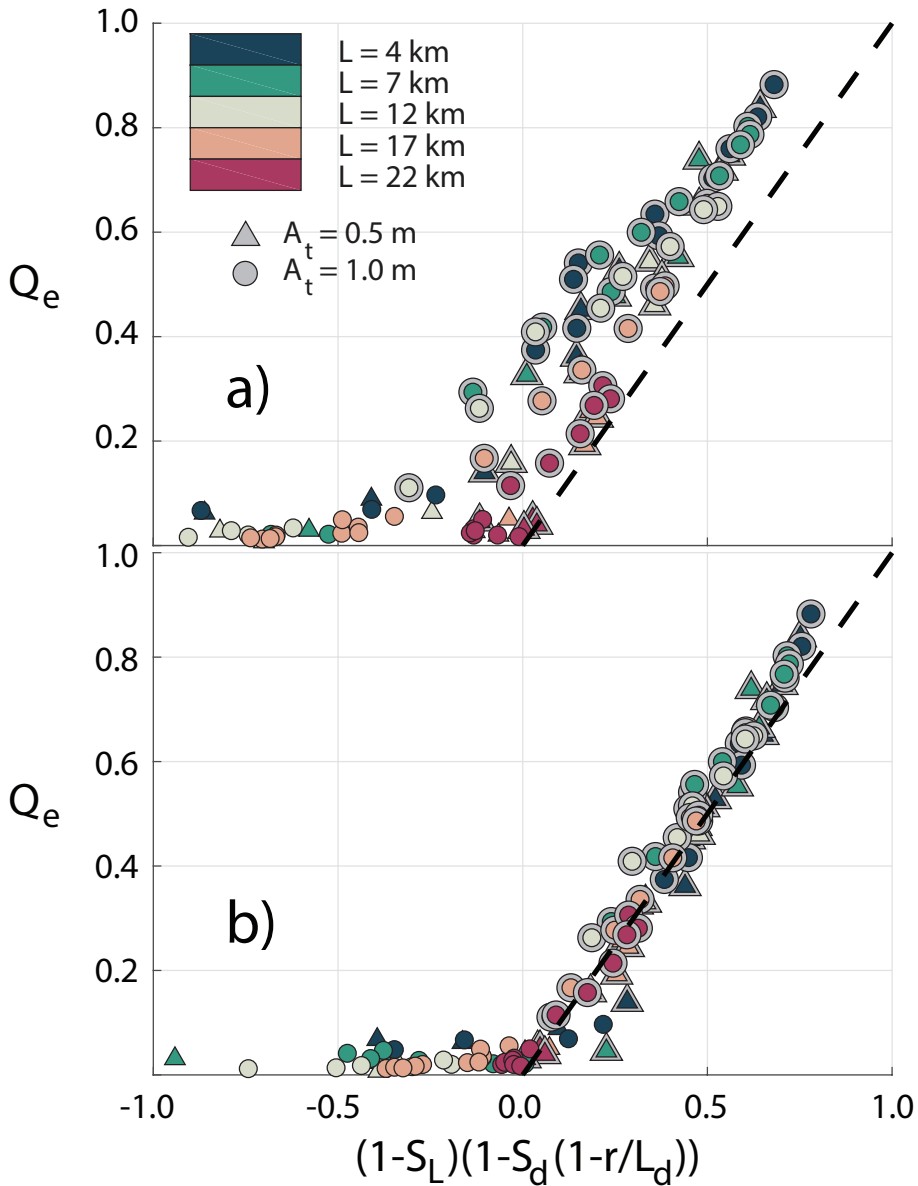

**Figure 18.** The effective transport $Q_e$ from the simulations (Eq. 19) plotted against the effective transport resulting from the simple kinematic model (Eq. 31) for a) $r/L_d = 0$ and b) $r/L_d = 0.4$. The dashed line indicates $Q_e = q_e$.

position $y$ along the strait. The equation governing the flow is,

$$\frac{\partial v}{\partial t} + v\frac{\partial v}{\partial y} = -g\frac{\partial \eta}{\partial y}, \tag{32}$$

where $\eta$ is the surface elevation. We have ignored friction, which is included by Garrett and Cummins (2005), because analysis of our model results show that it can be ignored in the momentum balance outside the viscous boundary layer. Volume conser-





vation implies that the volume flux $Q = Av$ cannot vary along the channel. Therefore, the along-strait variation of $v$ depends on the variation of cross-strait area $A$, and this sets a restriction on how the non-linear advection term can balance the pressure force. Integrating Eq. 32 over the full strait length leads to

$$c\frac{dQ}{dt} + \frac{1}{2}(v^2|_{y_e} - v^2|_0) = -g(\eta|_{y_e} - \eta|_0)$$

$$c = \int_0^{y_e} \frac{1}{A}dy. \tag{33}$$

Here $|_{y_e}$ and $|_0$ denotes strait exit and entrance respectively. From the integrated equation it is clear that it is only the linear advection term that increases with strait length, while the two other terms are given by the difference between values at the exit and entrance of the strait. Therefore, the flow dynamics becomes more linear as the strait length increases, and since $c$ increases with strait length, $dQ/dt$ will decrease as long as $\eta|_{y_e} - \eta|_0$ remains unchanged. Our model setup is designed such

that the difference in surface elevation across the strait is set by the tidal wave propagating around the peninsula and not by the strait flow. Therefore, $dQ/dt$ will decrease with strait length and this leads to a decreasing $U$ because the time period of acceleration is unchanged. Smaller $U$ leads to less dipole formations, which explains the results shown in Figure 3.

### 8.2 Dipole formation and flow separation

The dipole propagation velocity depends on the strength of the vortices set by their vorticity, and it is important to understand

how the vorticity is generated. Wells and van Heijst (2003) assumes that the vorticity is generated in the viscous boundary layer and injected into the vortices formed at the point of flow separation. Afanasyev (2006) introduces the "startup time", which is the time when the dipole starts propagating after an initial growth period being fed by the jet. Our simulations show a somewhat different picture. The dipole starts moving as soon as it is formed, and we see no initial period of growth. The dipole is formed at separation time (Figure 8), and before this we see no sign of vortices in the vorticity field (illustrated by Figure

5, upper panel). From the dipole tracking, we see that the dipoles start propagating immediately after they are formed (Figure 14).

The dipole formation is associated with a maximum in time of the absolute value of vorticity (Figure 9). This is an interesting phenomena and the question is whether the vorticity is a consequence of separation or if it plays an active role in causing the separation. Our results suggests that there is a buildup of vorticity before separation (Figure 9), which indicates that the vorticity

plays an active role in the separation process. The decrease in vorticity after separation might be connected to the roll-up of the velocity front creating the initial vortices. We see, from our simulations, that the core radius of the vortices increase and the maximum vorticity decreases with time. Assuming it would take time to build up the vorticity before another vortex is formed fits with the picture of maximum absolute value of vorticity occuring at separation time. Buildup and shedding of vorticity is also observed to be important role in controlling the separation point location of the flow around a wind turbine blade (Melius

et al., 2018).

The velocity front rolls-up immediately after separation and creates the dipole vortices (Figure 8). That the separated velocity front rolls up into a vortex is commonly observed in studies of flow separation (Délery, 2013), and that the velocity front is





the origin of the vorticity was also proposed by Kashiwai (1984a, b). During a time $T_*$ flow separation creates a velocity front of length $UT_*$ and the velocity difference across the front is $U$. Using Stokes theorem on an area enclosing the front we find

that the circulation of the front is $\Gamma \simeq U^2 T_*$ (Kashiwai, 1984b). Using this together with Eq. 17 and 18 the timescale $T_*$ can be expressed as

$$T_* = \frac{a\pi}{U}. \tag{34}$$

In our simulations, $U$ varies between 1 and 4 m/s, and the initial core radius $a$ is about 100 m for all simulations (Figure 11. This gives a timescale $T_*$ between one and five minutes. Thus, the initial vortices, making up a self propagating dipole,

is created within one to five minutes after flow separation. Vorticity is injected into the dipole also after separation and the circulation in the dipole increases. However, the order of magnitude of the total increase in the circulation is roughly similar to the circulation in the initial vortices. Therefore, the circulation of the dipole is well below the maximum possible given by $\Gamma_{max} \approx U^2(\frac{T}{2} - T_s)$, which occurs when all vorticity created in the separated velocity front is injected into the dipole.

The initial core radius of the dipole vortices (Figure 11) are very close to the smallest scale that can be resolved by the

model, which has a resolution of 50 m in the area of flow separation. With a finer resolution in the area of vortex formation, it is likely that the core radius of the vortices would be even smaller. This would probably lead to smaller and stronger vortices, as observed by Hutschenreuter et al. (2019). The viscous boundary layer on the other hand, would probably not see significant changes as this is well resolved. Following the same arguments as given by Eqs. 15 to 18, results in a smaller dipole propagation velocity because $\alpha$ will be smaller when the vortices have a smaller core radius. However, a finer resolution might lead to the

formation of several vortices along the velocity front created by separation. These vortices can merge and create a dipole similar to the dipoles created in our model. We believe that our model represents the behavior of dipoles in a realistic way, but studies of the actual formation of vortices and dipoles should be performed with finer resolution.

### 8.3 Dipole propagation velocity

As shown in Figures 14 and 15, Eq. 15 is a good representation of the dipole propagation velocity. However, Eq. 15 gives a

velocity that is twice as large as estimates obtained using Eq. 1. The aspect ratio of our simulated dipoles are mostly small ($\alpha \ll 1$) and the absolute maximum is about 0.5. For these aspect ratios Eq. 1 should be in good agreement with the simulated dipole velocities (Delbende and Rossi, 2009; Habibah et al., 2018), but instead the dipole propagation velocities are consistently twice as large. Recent work (Habibah et al., 2018) expresses the solution to the Navier Stokes equation in form of a power series in the aspect ratio. To first order the propagation velocity is given by our Equation 1, and a correction to this only appears

in the fifth order of the aspect ratio. In our case this correction should be small. Also, from Delbende and Rossi (2009) it appears that the propagation velocity actually decreases for increasing aspect ratio. Equation 1 gives the propagation velocity of a dipole moving in a non-moving ocean with no external forces acting on the dipole. These approximations are probably not valid in a tidal strait, where there is a strong background flow and vorticity and momentum are injected into the dipole by the tidal jet. We suspect that this is the reason for the discrepancy between Eq. 1 and the tracked dipole velocities.





A derivation of propagation velocity for a dipole connected to a jet is presented by Afanasyev (2006). The budget of volume and momentum in the dipole leads to a dipole velocity equal to half the jet velocity, in good agreement with observations. Afanasyev (2006) investigated a steady jet, but the mechanisms of momentum input from the jet to the dipole will apply also in our case of an oscillating tidal jet. We don't know the aspect ratio of the dipole studied by Afanasyev (2006), but it is not unlikely that it is around 0.5 and that his result therefore is in agreement with our result given by Eq. 18. Equations 15 and 18,

does not have a clear theoretical basis, but shows good fit to our large ensemble of numerical simulations. Further studies of dipoles formed in tidal straits are needed to fully understand the propagation of these dipoles.

## 8.4    A comparison to the Strouhal number

The Strouhal number (Eq. 2) is often used as a parameter describing the water exchange through a tidal strait by self-propagating dipoles(Wells and van Heijst, 2003; Kashiwai, 1984a). The dipole transport parameter $S_d$ (Eq. 24) is derived

in a similar way as Wells and van Heijst (2003) and Kashiwai (1984a) derived the Strouhal number. Comparing $S_d$ and $S_t$ we see that

$$S_d \approx \frac{S_t}{S_{tc}}, \tag{35}$$

where $S_{tc}$ represents the threshold value where $Q_e$ starts to increase from zero for smaller values of $S_t$ (Figure 19). Then in studies where strait length is constant (Kashiwai, 1984a; Wells and van Heijst, 2003; Nicolau del Roure et al., 2009), $S_t$ will

work well as a parameter for effective transport. One of the motivations for this study is that we found it difficult to understand the physics behind $S_t$ being a parameter for tracer transport through a tidal strait. Now, we see that $S_t$ is similar to $S_d$ which is a good parameter for effective transport if strait length is not varied (Figure 16a).

## 9    Summary and conclusion

In this study, we have performed a total of 164 numerical simulations in an ideal tidal strait, investigating flow separation,

dipole formation and water exchange for different widths and lengths of the strait. We show that dipoles are formed and start propagating at the time of flow separation. The vorticity of the dipole vortices originates from the velocity front created by flow separation. The simulated dipole propagation velocity is twice as large as the propagation velocity derived for vortex pairs with no background flow (Lamb, 1916; Delbende and Rossi, 2009; Habibah et al., 2018) (Eq. 1). This is probably caused by injection of momentum into the dipole by the tidal jet (Afanasyev, 2006).

We derive two parameters $S_d$ and $S_L$. $S_d$ (Eq. 24) is given by the ratio between sink radius and distance travelled by the dipole, while $S_L$ (Eq. 25) is given by the ratio between strait length and tidal excursion. For $S_L > 1$, the tracer will be contained within the strait through the whole tidal cycle and net transport is zero. For $S_d > 1$, the center of the dipole will be within the sink radius when the flow turns at $t = T/2$. However, since the dipole is of finite size a part of the dipole may still be outside the sink radius and escape the return flow causing net tracer transport. From a simple kinematic model we show that the effective

tracer transport can be expressed by $S_d$, $S_L$ and a parameter representing the dipole size ($r/L_d$, (Eq. 31). $1 - r/L_d$ acts as a

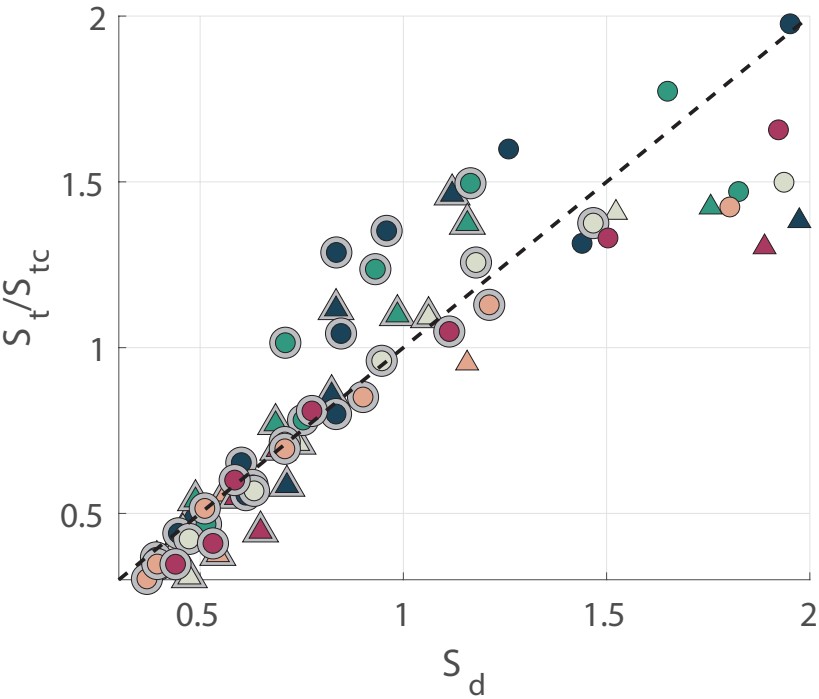

**Figure 19.** The Strouhal number $S_t$ dividen by its threshold value $S_{tc}$ plotted against $S_d$.

weight to $S_d$. Setting the value of $r/L_d$ such that effective transports are zero for values of the weighted $S_d$ larger than one, gives a remarkable good fit between the simple kinematic model and the numerical simulations (Figure 18).

The kinematic model (Eq. 19) provides an understanding of the processes creating a net tracer transport through a tidal strait. In our idealized straits, the sink radius is described by a half circle, the coastline curvature at the strait exit is kept constant and the strait is of uniform width. Along an irregular coast in the real world this will be different, but the physical processes will still be valid. An interested continuation of this study will be to derive $S_d$, $S_L$ and $r/L_D$ for a real coastline and investigate how well we can describe net tidal transports through straits.


*Code availability.* Model code is available at http://fvcom.smast.umassd.edu/fvcom/

*Author contributions.* Both authors have contributed equally





*Competing interests.*   No competing interest are present

*Acknowledgements.*   We thank P. E. Isachsen for constructuve scientific discussions and comments on the manuscript. E. Børve is funded by VISTA – a basic research program in collaboration between The Norwegian Academy of Science and Letters, and Equinor (project no. 6168). This work was supported by the Research Council of Norway (project no. 308796).





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
