# Peer review of "Flow separation, dipole formation and water exchange through tidal straits"

_Ocean Science, 2021_

## Author Comment (AC1)

**Sensitivity to mesh resolution: Figures**

June 1, 2021

[Figure]

Figure 1: The core radius (m) of the dipole vortices at $t = T_s$.

[Figure]

Figure 2: Time of flow separation, $T_s$, (hours)

[Figure]

Figure 3: The maximum vorticity at the strait exit at a function of time (hours) for the three example experiments. The time $T_s$ is marked with a triangle on each curve. This figure is the same as Figure 9 in the manuscript, but created from the 10m resolution simulations.

[Figure]

Figure 4: Velocity amplitude (m/s) in the strait

[Figure]

Figure 5: Tracked dipole velocitites (m/s).

[Figure]

Figure 6: Effective transports

---

## Author Response (AR2)

**Response to reviewers**

August 31, 2021

**1 Reviewer 1**

**The manuscript investigates formation and propagation of dipoles at tidal straits via numerical simulation. The authors derived equations to calculate dipole propagation velocity and to estimate net exchange through the strait. These equations show good agreements with the simulation results. This study advances the knowledge on transport mechanisms at tidal straits. The topic matches the scope of Ocean Science. It could be used as a basis for further studies on more realistic and more complex strait geometry and flow conditions. Thus, this manuscript is recommended to be accepted after some minor revisions listed below.**

    **- Line 234: Here r is defined as the distance from the vortex center, a is the vortex radius. However, in Fig. 17 and Eq. 31, if I understand it right, r becomes the vortex radius. I recommend to make the variable names consistent.**

    - In Fig. 17 (now Fig. 18) and the discussion around Eq. 31, we have now changed the variable name from $r$ to $r_d$. This should avoid any confusion with the $r$ in Eq. 11, which represents distance from vortex center.

    **- Figure 14: The tracked velocity is mostly lower than the theoretical values in (a), but higher than the theoretical values in (b). What might be the reasons?**

    *It is hard to know the exact answer to this question, and it could be a coincidence since Figure 14 (now Figure 15) only shows two examples. In Figure 16a, the theoretical dipole propagation velocity is compared to the tracked velocities for all simulations. There we see that on average they are abut equal. Sometimes the theoretical value is higher and sometimes it is lower. So, we conclude that on average there is close agreement, but we cannot explain the reason causing the differences in each simulation.*

    - Our work is about finding the first order physics. We see that there is an agreement between theoretical and tracked dipole velocities, but we cannot explain the differences in each simulation. Therefore we do not include a discussion on this in the manuscript.

    **- Line 374: The authors should provide, or at least discuss the valid range of Eq. 31. By ignoring the high order terms, Eq. 31 could be inaccurate when r is large (how to define "large"?). On the other hand, r=0 is not a realistic situation either. Moreover, r/Ld seems to be unknown before numerical simulations are completed. So the question is, given so many limitations, how useful is Eq. 31 in estimating the net transport? Why not simply calculate net transport through numerical modeling? I think the authors should provide more explanation on the significance and the applicability of the kinematic model.**

    *Equation 31 is important because it provides understanding of the processes. The agreement with the simulation results strongly indicates that the kinematic model contains the main processes at play. So, the main point of equation 31 is to provide understanding and not estimate transport. However, it can also be used to make rough estimates of transport if no simulations are present. Estimates on transport can be made*

*from the strait velocity and geometry, estimating dipole propagation velocity from Equation 18. This requires an estimate on the aspect ratio of the vortices, which largely depends on the strait width.*

*The main simplification leading to Equation 31 is that we estimate the fraction of the tracer inside the sink radius by assuming the dipole and jet has the form of a rectangle. This is of course not true, but it gives a simple expression for the effective transport. The improvement in the agreement with simulations resulting from adjusting r/Ld tells us that it is important to include in the model that only a fraction of the dipole escapes the return flow.*

*When it comes to ignoring the higher order terms, we see when going through this again, that this is not necessary to do. In the manuscript we use a constant value of r/Ld=0.4, for all simulations, which corresponds to (1-r/Ld) = 0.6. If not ignoring the higher order terms the expression for qe (Equation 31) becomes*

$$qe = (1 - SL) * (1 - Sd/(1 + r/Ld)).$$

*Setting 1/(1+r/Ld)=0.6, will give the exact same fit to the simulation data as in the manuscript. So, ignoring the higher order terms actually has no effect, and including all terms also gives a rather simple expression for qe.*

- In the revised manuscript we use the full expression for *qe* without ignoring the higher order terms. Thanks for setting focus on this and apologize for making this a little too complicated in the manuscript.

**- Figure 19: This figure shows that Sd and St are similar, so why do we need Sd? The authors state that St is "difficult to understand", but this is a very subjective statement. I encourage the authors to further explain the difference between Sd and St, as well as the advantage of using Sd (rather than St).**

*The point of this work is not to come up with a parameter that works better than the Strouhal number. The point of showing that Sd is similar to St is that the understanding provided by this work can also be used to understand St.*

*St is more like a scaling, using values of strait velocity, strait width and tidal period, while Sd comes from a kinematic model of tracer transport. However, it is hard to know which value of St which will give net transport. Figure 19 (in the old manuscript) shows Sd plotted against St/Stc, where Stc is the threshold value of St, such that St<Stc is gives non-zero tracer transport. In making the plot we have picked Stc from the simulated tracer transports. A threshold value of 0.13 is given by several authors, but this is likely to change with geometry. Then Sd is a simpler parameter, because it depends on sink radius and dipole travel distance. If these can be estimated, we know that Sd<1 may give transport if also SL<1.*

We have removed Fig. 19 and the comparison of Sd and St in the revised manuscript. With the added discussion regarding the mesh discretization, we felt the need to remove this Figure to keep the manuscript at a reasonable length.

**Overall, the language used in this manuscript is a little bit verbose. Unfortunately, as a non-native speaker, I cannot provide detailed recommendations on grammar. I suggest the authors try to make the language more succinct. A few typos I caught are:**

**- Line 386: Long straits produce less dipoles than "short" straits?**

**- Line 414: Our results "suggest" that . . . ?**

**- Line 460: Equation 15 and 18 "do" not have. . . ?**

We have done our best to make the language more succinct. The three typos listed by the reviewer are fixed

**The figures are generally informative. Some suggestions are:**

**- Figure 2: The figure labels "a" and "b" look similar to the variable names. The authors might consider separating them. For example, put labels outside the box, add parenthesis or change fonts.**

We have added paranthesis around the figure labels in all figures where lables are present.

**- Figure 16, 18 and 19: Some markers are with grey halos and some are not. The difference between these two types of markers is not explained in the caption or in the legend.**

The halos mark values from simulations where self-propagating dipoles are present. An explanation of this is now added to the figure captions.

**2 Reviewer 2**

**General overview**

**This is an interesting contribution that is within the remit of the Ocean Science journal. The authors discuss the dynamics of vortex dipole formation due to tidal flow. In general, it is an ambitious contribution that touches on a range of physics expected through the flow separation effects triggered through tidal straits**

*Many thanks for good and challenging comments from the reviewer. We have had special focus on investigating the effect of mesh resolution and done some new simulations with finer resolution. This has led to interesting results and deeper insight into the processes.*

**Specific comments**

**- On the definition of the case study - An idealised setup is explored and yet some practical elements seem to unnecessarily be included. For example, why would Coriolis effects be included in this case? Elements such as symmetry are also not exploited which could be interesting.**

*The Coriolis effect is vital for our setup to work, as the forcing comes from a Kelvin wave propagating with the coast to the right. A Kelvin wave cannot exist without the Coriolis effect. We have chosen this setup for two reasons: 1) It is an idealized model of the tides through straits in the Lofoten peninsula in Norway. 2) We wanted a setup where the pressure difference across the strait is not dependent on the flow through the strait. In our setup the pressure difference is mostly caused by the phase difference between the northern and southern part of the peninsula. This is setup by the Kelvin wave propagating around the peninsula, and less dependent on the flow through the strait. This is in contrast to, for instance, the entrance into a fjord, where the water flow through the entrance determines the water level in the fjord. The idea is that for theoretical considerations we can assume that the pressure difference across the strait is independent on strait flow. This is important when discussing friction and strait length (see below), where we can assume that the pressure difference over the strait is independent of the tidal velocity in the strait.*

The model is an idealized representation of the Lofoten peninsula in northern Norway, and this is now written into the manuscript (line 117-118). That the tidal wave is central to the setup is explained in the manuscript (see e.g line 115-117). The only forcing is the Kelvin wave (Eq. 9) which includes the Rossby radius in the expression. We hope that this is sufficient for a reader to properly understand the setup of the model and the inclusion of the Coriolis parameter. About symmetry: the Coriolis parameter probably leads to asymmetry across some of the widest straits. This would be an interesting topic to follow, but this is outside the scope of this manuscript, and we have therefore not included any exploitation of this.

**- "The water column is divided into two layers in the vertical" - does this provide sufficient vertical resolution? The depth the authors use could be used to argue that frictional effects are not so significant , (which is still not fully accurate as indicated later), but surely the flow separation (particularly in constricted straits) will lead to upwelling that will affect the vortex evolution. How is vorticity calculated? is it averaged across the 2 layers or is the top layer considered in the analysis**

*We regard this as a 2D barotropic study. The actual reason for having two vertical levels in the model is that our tracer model did not work within a 2D model setup and the simplest solution to this was to include two levels. However, all our analysis is done using vertically averaged velocities.*

We have added some text in section 2.2 to explain that this is a 2D barotropic study and we use vertically averaged velocities in the analysis.

**- Mesh discretisation - The analysis is based on spatially defined metrics such as vorticity and circulation. These are extremely sensitive to the resolution of the model. The fine resolution close to**

the strait bounds can lead to higher vorticity peaks that will influence the conclusions of the study - while coarse resolution will dissipate the vortices. It is challenging to conduct this analysis using unstructured models, and some earlier studies included a mesh sensitivity, and a normalisation based on the element length to check whether the key dynamics are accurately preserved independently of the mesh resolution (see e.g. Vouriot et al, Env. Fl. Mech., 19, 328-348 (2019)). It is understood that the same issue was indicated in an earlier submission of this manuscript so this must be addressed. For example, Eq. 12 seems heavily affected by the mesh resolution and must proven that it is mesh independent.

*Using the circulation theorem, we argue that the total vorticity production is not very sensitive to resolution even though the vorticity is sensitive to resolution. Since dipole propagation velocity is dependent on circulation it will not be sensitive to mesh resolution, as long as the tidal velocity and flow separation is realistically represented. We have added section 8.1 where we discuss the effects of mesh discretization. To confirm our theoretical analysis, we have rerun seven of the simulations using 10 m resolution at the coast compared to 50m in the original simulations. Please read section 8.1 for details of our analysis*

We have added section 8.1 where we discuss the effects of mesh resolution.

**- Increasing length of the strait will increase the effects of friction, and thus influence u and Q through the strait. This seems to be the reason for the monopol/dipole trends of Fig.3. However, this is the reason why I suspect that Section 8.1 is invalid as friction becomes more and more important as the strait length increases.**

*Frictional effects will increase with increasing strait length, and this is also the case for the linear acceleration term. So if it is friction or linear acceleration depends on the relation between these two terms. This relation is not dependent on strait length. Here it is important to be aware that the pressure difference across the length of the strait is independent on depth due to the model setup (see first paragraph section 2.2). The relation between linear acceleration and friction then depends on depth, and in our case it is the linear acceleration which is the main cause of the length effects seen in Fig. 4. However, for a general discussion it is important to include friction*

We have rewritten the section about strait length. In the original manuscript the friction was not included in the discussion. Now the discussion is more general as friction is included. See section 8.2.

**- How was $\Gamma$ determined for each vortex? How was Ts estimated and is $u\theta$ the propagation velocity of the vortex?**

*$\Gamma$ was determined by the Lamb-Oseen equations (Equation 11 in the manuscript) fitted to the simulated vortices. Through the fitting, we find the core radius a, and the maximum vorticity. When these two variables are known $\Gamma$ can be determined from Equation 11b.*

*$u\theta$ (in Equation 11a) is the azimuthal velocity of the vortex. In the manuscript we determine Ts by finding the maximum (or minimum) vorticity within the strait exit. We see by visual inspection that this corresponds well with the time of separation. However, in the new simulation with finer resolution presented above, the maximum (minimum) vorticity method does not work, and Ts is then determined by visual inspection of the simulation results.*

That $T_s$ is found using the maximum vorticity is written in line 225. That this method does not work for the high resolution simulations is written in line 4156-420.

**3   Reviewer 3**

**The manuscript describes the generation, dynamics, and transport properties of a pair vortices through tidal straits, which is a key mechanisms in water exchanges. A set of 164 simulations with varying strait width, length, and tidal amplitude is investigated. Along with comments already posted by the two referees, a few more suggestions are provided below.**

**The paper is within the scope of Ocean Science.**

**The manuscript is well written and the physical mechanisms at play are extensively described. However, it somehow lacks of conciseness, and the reader can get lost in long descriptions that could be replaced, or complemented by illustrative sketches (or simulation outputs) for the sake of clarity. An example is the description of the origin of the asymmetry leading to net exchange (l18-30).**

*We have done our best to improve the language of the manuscript to make things more concise and clear. The specific comment about the text on the origin of the asymmetry is rewritten and we have added Fig 1 which is a simple sketch of the processes leading to asymmetry in the flow field.*

**Introduction:**

**The authors quote a few experimental studies. Recent work by Albagnac et al (i.e. A three-dimensional experimental investigation of the structure of the spanwise vortex generated by a shallow vortex dipole. (2014) Environmental Fluid Mechanics, vol. 14 (n° 5). pp. 957-970. ISSN 1567-7419, and later articles) provide quantitative description of the 3D dynamics of vortex dipoles, including in stratified environment.**

*We now briefly mention the paper by Albagnac in the introduction together with a paper by van Heijst (Shallow flows: 2D or not 2D?). These papers are cited to make the reader aware that there may be 3D effects, although we still choose to use a 2D approach*

**Sec 3: an illustration of the effect of tidal amplitude would have been interesting.**

*Figure 4 also shows the effect of tidal amplitude. Basically, lower amplitude gives lower velocities resulting in less dipoles being formed. The three examples shown in Figure 5-7 is chosen because they show three typical patterns summarizing what we see in all straits and for both tidal amplitudes used. In our opinion Figure 4 is a good illustration of the effect of tidal amplitude.*

We have not included more illustrations of tidal amplitude because we believe this is well represented in the paper through Fig. 4.

**Fig 13: S1 and S2 are not described in in the caption.**

We have changed this figure and replaced S1 and S2 with b1 and b2 representing the distance between the two dipole vortices at time t1 and t2.

**l412: phenomena - > phenomenon.**

This is fixed

**Conclusion: Multitidal forcings are often present at straits, an interesting perspective would be to investigate their effect on the overall dynamics and the impact on water exchanges.**

*Yes, multitidal forcing will impact the water exchange through tidal straits. We do not specifically focus on multitidal forcing here because this will widen the topic too much. However, multitidal forcing will basically lead to a spring-neap tidal cycle where the forcing amplitude will vary on a ~14 day period. We see that higher forcing leads to larger exchange, so for multitidal forcing the exchange will likely vary over a ~ 14 day period.* We have not included a discussion on multitidal forcing, because we see this as a little outside the scope for this paper.

**As suggested by the authors, comparisons with realistic configurations would be of interest to evaluate the parameters controlling the dipole dynamics. A relevant configuration could be the one of the Gibraltar strait, where the variety of finescale structures and the water exchange through the strait are actively studied combining LES simulations and sea campaigns (Numerical modelling of hydraulic control, solitary waves and primary instabilities in the Strait of Gibraltar, Hilt et al. Ocean Modelling, 2020).**

*We hope that our results will be used as a basis for realistic studies of tidal pumping. We are doing work on this now, and a paper is already submitted to Ocean Science (https://doi.org/10.5194/os-2021-41) and considers tidal transports in the Lofoten region in Northern Norway. The Gibraltar Strait would be an interesting study, but we will focus on the Norwegian coast as this is where we have funding to work. In the submitted paper we use a barotropic model of the tides in the regions and study tidal pumping through straits*

*and rectified transports around islands. A third paper presenting a 3D study of the same region is underway. Here we examine the role of tidal transports in a model that also includes atmospheric forcing and river runoff. In this work, we clearly see that multitidal forcing is important.*

**4   Additional changes to the manuscript**

In addition to the changes made in the response to reviewers, we discovered a few other errors when working through the manuscript. These are not serious, but we still report them here.

1. We added a missing $\rho_0$ in Eq. 7.

2. The fraction in Equation 13 was turned upside down. This is now corrected.

3. The original expression in Equation 23 should be multiplied by $\pi$ to be correct. This is now fixed.

4. In the expression for $S_L$ (Eq. 25) we have not used the same velocity scale as in the expression for $S_d$. $S_L$ uses the cross strait maximum velocity to assure that no transport occurs for $S_L > 1$. This is now corrected in Eq. 25 and in the text.

5. In our calculations of $S_d$ we found an error. $S_d$ then became a factor of $\sqrt{2}$ wrong. The expression for $S_d$ (Equation 24) is correct but our calculations had an error. We have now corrected this resulting in somewhat larger values for $S_d$ in Fig. 17. The correction is simply to multiply the old values by $\sqrt{2}$, and therefore conclusions are not affected.